# Barents-Kara sea-ice decline attributed to surface warming in the Gulf Stream

Yoko Yamagami [1✉], Masahiro Watanabe [2], Masato Mori [3] & Jun Ono [1]

Decline in winter sea-ice concentration (SIC) in the Barents-Kara Sea significantly impacts climate through increased heat release to the atmosphere. However, the past Barents-Kara SIC decrease rate is underestimated in the majority of Coupled Model Intercomparison Project Phase 6 (CMIP6) climate models. Here we show that climate model simulations can reproduce the Barents-Kara SIC trend for 1970–2017 when sea surface temperature (SST) variability in the Gulf Stream region is constrained by observations. The constrained warming of the Gulf Stream strengthens ocean heat transport to the Barents-Kara Sea that enhances the SIC decline. The linear trends between the SIC and SST are highly correlated in the CMIP6 ensemble, suggesting that the externally forced component of the Gulf Stream SST increase explains up to 56% of the forced Barents-Kara SIC trend. Therefore, future warming of the Gulf Stream can be an essential pacemaker of the SIC decline.

[1] Japan Agency for Marine-Earth Science and Technology, Yokohama, Japan. [2] Atmosphere and Ocean Research Institute, University of Tokyo, Kashiwa, Japan. [3] Research Institute for Applied Mechanics, Kyushu University, Kasuga, Japan. ✉email: y.yamagami@jamstec.go.jp

There is observational and modeling evidence that the recent retreat of Arctic sea ice has been driven by anthropogenic greenhouse gas emissions[1–3], and climate projections using multiple global climate models (GCMs) suggest that near ice-free conditions will emerge in the Arctic Ocean in September by the middle of this century[4–8]. Overall, the reproducibility of the distribution and past variations of Arctic sea ice has much improved in the recent GCM generation[1–3,8,9]. However, the CMIP6 GCMs still have difficulties in reproducing sea ice in more localized region such as the Barents-Kara Sea (Fig. 1d).

In past decades, even in winter (December, January, and February; DJF), when sea-ice formation is largest, sea ice has pronouncedly decreased in the Barents-Kara Sea[4,10–12] (Supplementary Fig. 1). The winter SIC anomalies strongly affect Arctic climate variability by changing the heat exchange between the warm ocean and cold atmosphere, and may even remotely influence the mid-latitude surface air temperature as suggested in recent studies[13–17]. However, the CMIP6 multi-model mean underestimates the winter Barents-Kara SIC negative trend (20°–70° E and 65°–85° N; Fig. 1d).

At interannual time scales, several drivers of Barents-Kara sea-ice area variability have been suggested by observational and modeling studies. The winter sea-ice area tends to be lower than usual when warm water intrudes from the North Atlantic, which is related to a process called "Atlantification"[18–22]. This occurs via the anomalous advection of warm and saline water by the North Atlantic Current, which transports heat from the North Atlantic subpolar region and thereby affects the oceanic heat content in the Norwegian Sea and the Barents Sea[21,23–25]. Additionally, it has also been suggested that a meridional shift of the SST front along the Gulf Stream can excite atmospheric waves that result in

a SIC reduction by enhancing the meridional atmospheric heat transport into the Barents-Kara Sea[25–27]. Although the direct impact of ocean heat transport from the Gulf Stream on Barents-Kara Sea is still unclear, fluctuations in meridional heat transport from the North Atlantic, either by the ocean or the atmosphere, can be a source of interannual variability of the Barents-Kara SIC[11,25,28].

Our current knowledge of the interannual variability in Barents-Kara SIC described above naturally leads us to hypothesize that the less negative trend of winter Barents-Kara SIC (Methods) in CMIP6 multi-model mean (Fig. 1d) is related to an inaccurate simulation of the North Atlantic SST, especially around the Gulf Stream. To verify this hypothesis, we investigated an ensemble of historical simulations in which SST anomalies in the Gulf Stream region were constrained by observations (so-called pacemaker experiment). This pacemaker experiment can reproduce the SIC trend averaged over the entire Barents-Kara Sea for 1970–2017 better than the CMIP6 historical simulation products, and shows that more than half of the SIC trend can be explained by the ocean surface warming in the Gulf Stream region. By comparing the linear trends in SIC and SST, physical processes linking these trends are identified and discussed.

## Results

**Ensemble North Atlantic–Global Atmosphere experiment.** Two sets of experiments for 1970–2017 are conducted using the Model for Interdisciplinary Research on Climate version 6 (MIROC6[29]), which is one of global climate models participating in CMIP6[30]. One involves historical experiments with a 50-member ensemble (denoted as HIST) based on the configuration of the CMIP6 historical runs. The other set of experiments is similar to HIST,

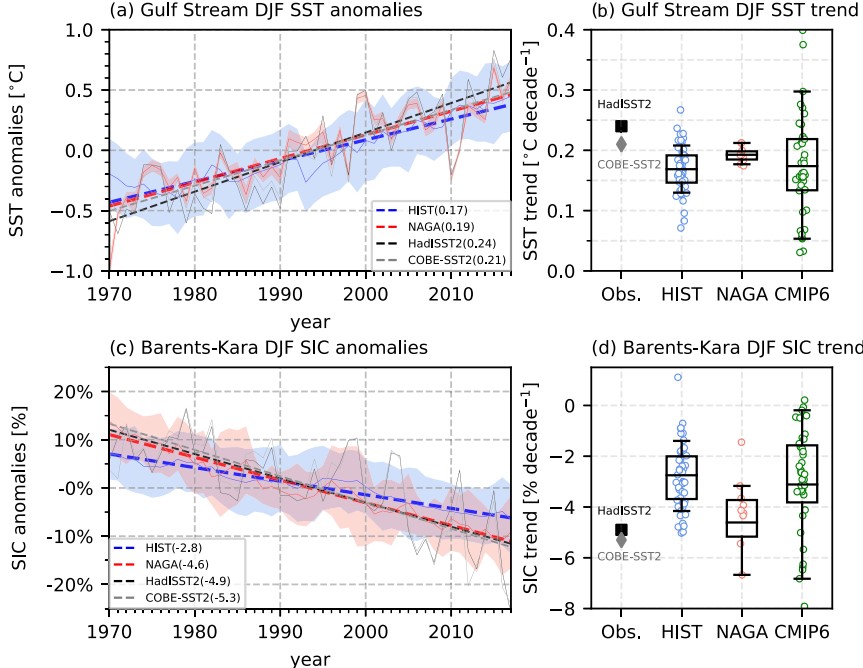

**Fig. 1 Observed and simulated time series and linear trends of winter SST in the Gulf Stream and SIC in the Barents-Kara Sea for 1970–2017. a** DJF mean SST anomalies (thin lines) and linear trends (dashed lines) averaged over the Gulf Stream region (30°–80° W, 30°–50° N; the region is indicated with a box in Fig. 4) based on the observation (HadISST2; black, COBE-SST2; gray) and the ensemble means of 10 members for HIST (blue) and NAGA (red). Shading indicates one standard deviation of ensemble members in both experiments. Linear trend per decade for each data set is shown in the legend. Ten members for HIST are estimated using the Monte Carlo method (Methods). **b** Box-whisker plots of DJF Gulf Stream SST trends for 10 equivalent members (Methods) in HIST (blue), 10 members in NAGA (red), and 39 members in CMIP6 (green). The boxes extend from the 25% to 75% values of the data, with a line at the ensemble mean. The whiskers denote the range from 5% to 95% of the data. Circles indicate all members for HIST (blue), NAGA (red), and CMIP6 (green). Black and gray markers indicate HadISST2 and COBE-SST2, respectively. **c, d** As in **a** and **b**, but for Barents-Kara SIC (20°–70° E, 65°–85° N; the region is indicated with a box in Fig. 2a, b).

except that modeled SST anomalies are restored to observed anomalies in the Gulf Stream region (30°–80° W, 30°–50° N); this 10-member ensemble is called the North Atlantic–Global Atmosphere (NAGA) experiment (see Methods).

The HIST ensemble mean for the winter (December-January-February; DJF) Gulf Stream SST trend for 1970–2017 is 0.17 ± 0.04 °C decade$^{-1}$ (where the range denotes one standard deviation), and this value is similar to the CMIP6 multimodel mean (0.17 ± 0.08 °C decade$^{-1}$), but considerably smaller than observed values (0.24 °C decade$^{-1}$ by HadISST2 and 0.21 °C decade$^{-1}$ by COBE-SST2) (Fig. 1a, b and Supplementary Fig. 2a). Similarly, the winter Barents-Kara SIC in HIST exhibits a decreasing trend of −2.8 ± 1.4% decade$^{-1}$, which is close to the CMIP6 multimodel mean trend (−3.1 ± 2.3% decade$^{-1}$) but smaller than the observations from HadISST2 (−4.9% decade$^{-1}$) and COBE-SST2 (−5.3% decade$^{-1}$) (Fig. 1c, d and Supplementary Fig. 2b). In NAGA, the winter Gulf Stream SST trend is very similar to the observations by the experimental setting, and furthermore, the winter SIC trend in the Barents-Kara Sea is more negative than that in HIST by 64 % (−4.6 ± 2.0% decade$^{-1}$) and comparable to the observations (Fig. 1b, d).

The observationally constrained SST anomalies in NAGA act to regulate the SST variability outside of the Gulf Stream (Supplementary Fig. 3). In the North Atlantic, the ratio of ensemble-mean variance to total variance in the ensemble, analogous to the signal-to-noise ratio (Methods), is approximately 0.2 for HIST, which indicates the minor role of the external forcing in producing interannual SST variations. In NAGA, the ratio is nearly unity in the Gulf Stream region and exceeds 0.4 in the Barents-Kara and Norwegian Seas. These values suggest that the constrained SST variability in the Gulf Stream influences temperatures in the Barents-Kara and Norwegian Seas. A similar ratio calculated for sea level pressure displays a negligible difference between HIST and NAGA, which indicates the limited impact of constrained SST variability on the atmosphere above. Thus, by constraining the Gulf Stream SST variability, the improved SIC trend in the Barents-Kara Sea is likely to result from oceanic processes.

**Processes responsible for the Barents-Kara sea-ice concentration decline.** Given the importance of links between the North Atlantic SST and the Barents-Kara SIC variation as suggested in the literature[25], the further upstream Gulf Stream SST can be the source of the ocean and/or atmospheric heat transport to the Barents Sea, which influence sea-ice decline there. The linear trends of surface ocean and lower atmosphere variables are compared between HIST and NAGA over the Barents-Kara Sea to determine whether atmospheric or oceanic variability contributes to the decrease in SIC (Fig. 2). Differences in surface wind trends between HIST and NAGA appear to cause sea ice to retreat more poleward in NAGA, but the trends are not statistically significant (Fig. 2c, d). Moreover, sea-ice drift trends in NAGA and HIST respond to surface ocean circulation changes rather than wind changes (Fig. 2a–f), suggesting the differences in sea-ice loss between NAGA and HIST have an oceanic origin rather than atmospheric one. While, the spatial agreement of negative trends in NAGA between the SIC and surface salinity indicates the decrease in sea-ice formation, which leads to the negative trend of surface heat fluxes to the ocean (i.e., heat release to the atmosphere) (Supplementary Fig. 4a, c). Hence, the atmospheric surface warming trend (Fig. 2c) results from sea-ice loss rather than the cause. The same is true in the difference between NAGA and HIST. The heat release difference is due to the lower sea-ice formation (i.e., the lower SIC), which is implied by the more negative trends of surface salinity flux in NAGA

(Supplementary Fig. 4c, d). Therefore, the improved SIC trend in the Barents-Kara Sea results from the SST warming in NAGA compared to HIST (Fig. 2e, f), possibly driven by oceanic heat transport from the North Atlantic domain.

To show the dynamical link between the North Atlantic and the Barents-Kara Sea, we examine the transects of ocean potential temperature and horizontal heat transport to the north of 70°N (Fig. 3). We find that the poleward heat transport from the North Atlantic (measured at two different transects) increases more in NAGA than HIST, which leads to the faster reduction of Barents-Kara SIC in NAGA. This increase is larger at 70°N compared to the Barents Sea Opening section, where the increase is localized to the northern part of the section (Fig. 3). Indeed, the heat content in the upper 345 m of the ocean increases in NAGA compared to that in HIST (Supplementary Fig. 5).

Since water temperature and horizontal heat transport are vertically uniform in the upper layer (Fig. 3a, b), the subsurface horizontal heat flux trends at 54 m depth are investigated. The horizontal heat transport across the Barents Sea Opening section increases and reaches the Barents-Kara Sea in NAGA (Supplementary Fig. 6). In Barents-Kara Sea, a large difference between HIST and NAGA is found at 75° N, 15°−30°E, leading to higher surface warming and a larger SIC decrease in NAGA. To understand the cause of the differences in the ocean heat transports between HIST and NAGA, the horizontal subsurface heat flux trends are decomposed into the contributions of trends in temperature, ocean current velocity, and their covariability (Methods). In both experiments, the increase in the eastward horizontal heat flux is explained by temperature-induced trends in the center of the Barents-Kara Sea (Supplementary Fig. 6e, f). However, the contribution of velocity and covariability trends in the Norwegian Sea and the northern Barents Sea is larger in NAGA than HIST (Supplementary Fig. 6c–h), suggesting an essential contribution to the ocean circulation.

**Ocean circulation change.** To understand the mechanism of ocean circulation that enhanced the heat transport to the Barents-Kara Sea, we investigate linear trends of the horizontal velocity and potential temperature vertically averaged within the surface mixed layer in the region between the Gulf Stream and the Norwegian Sea (Fig. 4). In NAGA, both magnitudes of the velocity and temperature trends are larger than those in HIST and suggest that surface warming in the Gulf Stream region accompanies a cyclonic circulation that weakens the subpolar gyre (Fig. 4a, b). This slowdown trend of the Gulf Stream is consistent with observations[31]. In both HIST and NAGA, ocean dynamics contribute to surface temperature warming in the Norwegian Sea (Fig. 4e, f), but its contribution is larger in NAGA and extends to the downstream of the Gulf Stream (around 30°W and 50°N), which is consistent with the larger temperature warming in the Norwegian Sea (Fig. 4c, d). The positive ocean contribution trend is found along the acceleration trends of surface velocity, suggesting that poleward ocean heat transport strengthens (Fig. 4e, f). In NAGA, the positive temperature trend between the Gulf Stream and the Norwegian Sea is interrupted by a negative trend (Fig. 4c, d). This is due to enhanced cooling by surface heat fluxes in the subpolar region (Fig. 4g, h), which do not mean a break in poleward ocean heat transport.

To demonstrate how SST anomalies in the Gulf Stream induce the entire North Atlantic circulation response, we conducted an idealized 10-year experiment named NAGAc, in which the modeled SST anomaly is restored to the idealized positive SST anomalies (Methods) in the Gulf Stream region (Fig. 5a; Supplementary Fig. 7). The time evolution in NAGAc indicates that the Gulf Stream warming slowly propagates and reaches the Barents-Kara Sea after approximately 7 years (Fig. 5b–l). The

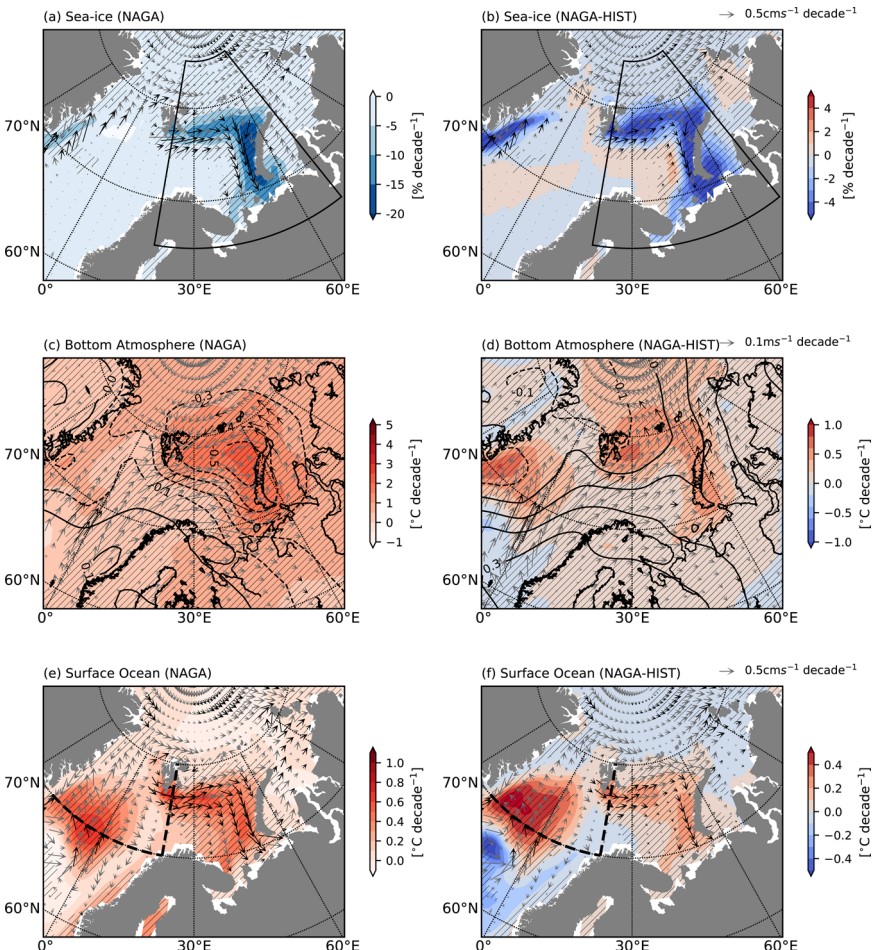

**Fig. 2 Simulated linear trends of the winter SIC, sea-ice drift, and surface atmospheric and oceanic circulation in the Barents-Kara Sea for 1970–2017 for the NAGA and HIST experiments.** Linear trends of DJF mean SIC [% decade$^{-1}$] (colors) and sea-ice drift [cm s$^{-1}$ decade$^{-1}$] (vectors) for **a** NAGA and **b** the difference (NAGA minus HIST). The hatching and black vectors in **a**, **b** indicate statistically significant linear trend for NAGA. **c**, **d** As in **a** and **b**, but for surface air temperature at 2 m height [°C decade$^{-1}$] (colors), 10 m wind [m s$^{-1}$ decade$^{-1}$] (vector), and sea level pressure [hPa decade$^{-1}$] (contours). The contour interval is 0.1. Solid (dashed) contours are positive (negative) trend. **e**, **f** As in **a** and **b**, but for SST [°C decade$^{-1}$] (colors) and surface ocean currents [cm s$^{-1}$ decade$^{-1}$] (vector). Black dashed lines indicate transects in which potential temperature and horizontal heat transports are investigated in Fig. 3.

temperature anomalies during the first three years develop within the subtropical gyre and the Gulf Stream region, and then propagate northeastward along the European coast. The warm temperature anomalies are partly advected into the Barents-Kara Sea, whereas the rest accumulates in the Norwegian Sea, leading to warming there. The Barents-Kara Sea warming 7 years after the Gulf Stream SST warming in NAGAc is consistent with the significant differences in the heat content in the Barents-Kara Sea between HIST and NAGA after 1977 (Supplementary Fig. 5).

**Barents-Kara sea-ice concentration decrease and Gulf Stream warming in CMIP6 climate models.** The robustness of the relationship between Gulf Stream warming and the Barents-Kara SIC decrease can be examined by analyzing multimodel historical CMIP6 simulations (Supplementary Table 1). We used a single member from 39 models for the period of 1970–2014 as the CMIP6 historical run ends in December 2014. The multimodel mean trends for both Gulf Stream SST and Barents-Kara SIC are weaker than the observed values, and a significant negative correlation ($r = -0.59$) is found between the linear SST and SIC trends across models (Fig. 6a).

Since the intermodel spread in 39 CMIP6 models contains differences in the forced signal and internal variability, we chose

14 models (including MIROC6) that have more than 10 ensemble members (Supplementary Table 2). In those models, the ensemble mean reflects the forced signal, and deviations from it represent internal variability. For 50 members of MIROC6 HIST, there is no significant correlation between the internally generated linear trends in SST and SIC ($r = -0.14$; Fig. 6b). Likewise, the SST-SIC correlation is insignificantly low in nine-thirteenths of the other models (Supplementary Fig. 8). In contrast, forced linear trends in 14 models as defined by the respective ensemble mean, exhibit a significant negative correlation ($r = -0.75$; Fig. 6c). This result indicates that the Gulf Stream warming driven by external forcing, plausibly due to increased GHGs, could explain 56% of the forced Barents-Kara SIC decrease for 1970–2014. Thus, forced warming of the Gulf Stream is an important process related to the Barents-Kara SIC decline over the past five decades. While, even though the SST trend of each NAGA member is constrained by the same observations, the variation of SIC trends between ensemble members is as large as that in HIST, suggesting that factors other than the Gulf Stream SST also contribute to SIC variability.

To clarify the sources of the inter-model spread of the linear trends, an empirical orthogonal function (EOF) analysis is performed for the Gulf Stream SST trends in 39 CMIP6 models

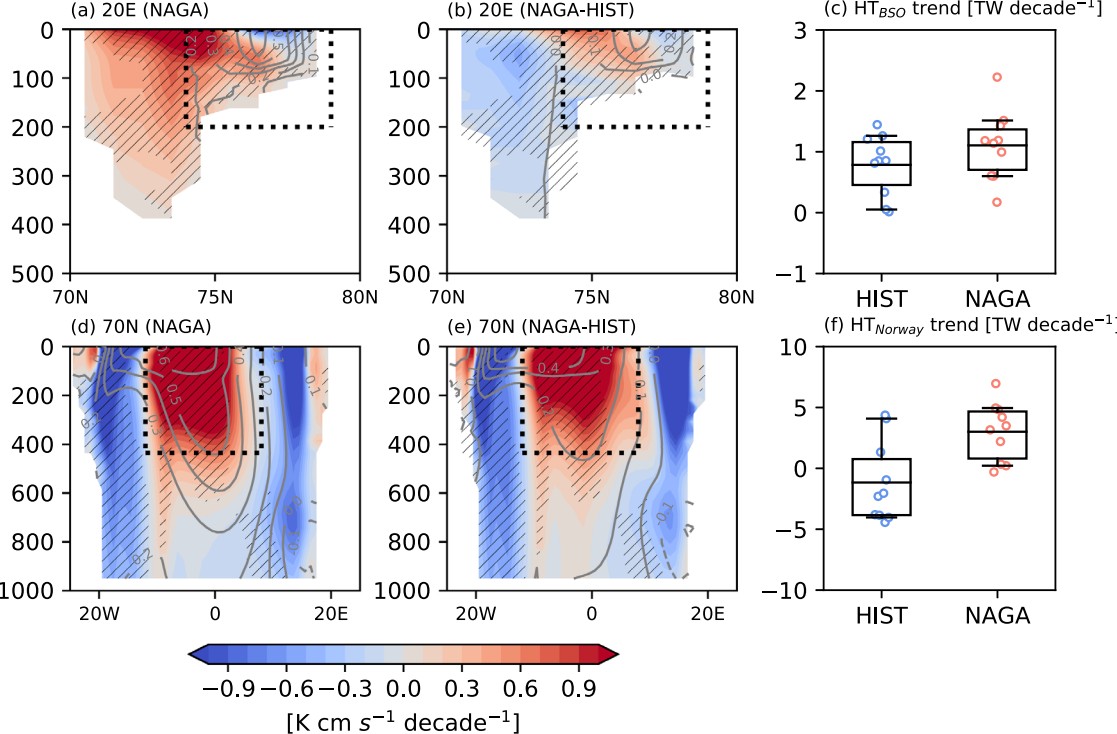

**Fig. 3 Linear trends of horizontal heat transport at the Barents-Sea Opening section and the Norwegian Sea. a, b** Simulated linear trends of DJF mean zonal heat flux [K cm s$^{-1}$decade$^{-1}$] (color) and potential temperature [°C decade$^{-1}$] (gray contours) in the Barents-Sea Opening section (20° E, 70° N to 80° N; black dashed lines in Fig. 2e, f) for **a** NAGA and **b** the difference between NAGA and HIST for 1970–2017. The hatching indicates statistically significant linear trend of heat flux for NAGA. The contour interval is 0.1 and the dashed line indicates negative. **c** Box-whisker plots of zonal heat transport trends for 10 members in HIST (blue) and NAGA (red). Heat transport is integrated within the black dashed boxes in **a** and **b** (74°N-79°N, surface-198m). The box extends from the 25% to 75% values of the data, with a line at the ensemble mean. The whiskers denote the range from 5% to 95% of the data. Circles indicate ten members for HIST (blue) and NAGA (red). **d–f** As in **a–c**, but for meridional heat transport trends at 70°N. Meridional heat transport trends in **f** are integrated within the boxes in **d** and **e** (12°W-9°E, surface-435m).

after the multimodel mean are removed to capture the inter-model anomalies. The leading EOF (accounting for 35.4% of the total variance) reveals that the maximum SST trend occurs in the northern Gulf Stream region; the larger the signal in a model, the larger the decrease in Barents-Kara SIC (Supplementary Fig. 9a, b).

When a similar analysis is performed by using CMIP6 models having more than 10 ensemble members, the intermodel variations in SST trends are dominant in the subpolar gyre, the Norwegian Sea, and Barents-Kara Sea (Supplementary Fig. 9c). The intermodel differences in the corresponding principal component measure well the differences in the ensemble-mean SST/SIC trends (Supplementary Fig. 9d). Furthermore, a similar analysis was separately applied to the ensemble means and deviations from it in each model. The leading EOF of the internally generated trends (i.e., deviations from ensemble mean) in all models represents the SST signal limited to the Gulf Stream region with small trends in the Barents-Kara SIC (Supplementary Fig. 9f). In contrast, the leading EOF for the forced SST trend (i.e., ensemble means) shows large signals over the subpolar gyre, the Norwegian Sea, and the Barents-Kara Sea, as well as SIC (Supplementary Fig. 9e), highlighting the importance of the forced SST trend.

Previous studies reported a relationship between Arctic sea-ice changes and Atlantic multidecadal variability (AMV)[32–34], and the SST trend in CMIP6 exhibits similarities to the SST anomaly pattern associated with AMV (Supplementary Fig. 7). Indeed, the linear trends for the Gulf Stream SST and the DJF AMV index (area-averaged SST; 0°–65° N, 0°–80° W) are well correlated ($r = 0.86$) based on the ensemble means of 14 CMIP6 models (Supplementary Fig. 10). Although the relative contributions of externally forced signals and internal variability to observed AMV are still controversial, our result is consistent with those studies that have reported externally driven AMV in past decades[35–37].

## Discussion

We demonstrated that the Barents-Kara SIC decrease and the Gulf Stream SST increase are tightly connected by the ocean circulation in the North Atlantic. This relationship is found both in a multi-member analysis performed with MIROC6 and in the CMIP6 multi-model ensemble. Although a connection between the North Atlantic SST and the Barents-Kara SIC at interannual timescales has been reported[25–27], our results go beyond this simple relationship and show that the surface warming long term trend in the Gulf Stream region is an important pacemaker of the Barents-Kara SIC decrease over the past five decades.

The model underestimation in the Gulf Stream warming is likely to explain the small SIC trend encompassed by the CMIP6 multimodel mean compared to observations. Uncertainty of the forced SST response in the Gulf Stream region in climate models is important, and the reason for the underestimation of the SST trend should be investigated in future studies. This study implies that reducing uncertainty of the SST response in the North Atlantic to anthropogenic GHGs and aerosol forcing may be crucial for simulating the Arctic sea-ice change.

## Methods

**Observational data**. For monthly SST and SIC, Centennial In Situ Observation-Based Estimates of the Variability of SST and Marine Meteorological Variables version2 (COBE-SST2)[38] and Hadley Centre Sea Ice and Sea Surface Temperature version 2 (HadISST2)[39] are used for 1970-2017.

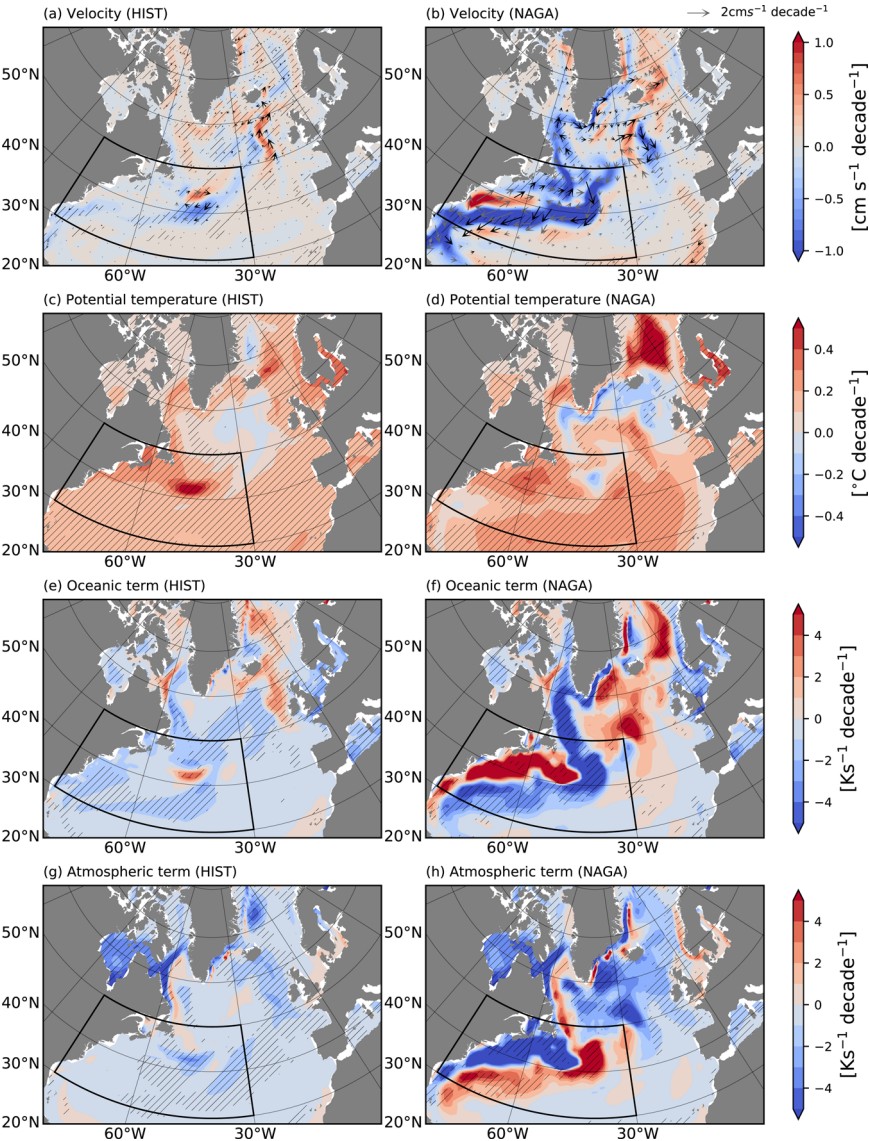

**Fig. 4 Linear trends of the potential temperature and horizontal velocity vertically-averaged within the surface mixed layer in the North Atlantic for NAGA and HIST.** Linear trends in the DJF mean horizontal current (vector) and velocity (colors) [cm s$^{-1}$ decade$^{-1}$] for **a** HIST and **b** NAGA. The horizontal velocity is vertically averaged within the surface mixed layer. The hatching and black vectors indicate statistically significant linear trend. The black solid box shows the area in which SST anomalies are restored to the observations in NAGA. **c–h** As in **a** and **b**, but for **c** and **d** for potential temperature [°C decade$^{-1}$], **e** and **f** oceanic contribution to the mixed layer temperature tendency [Ks$^{-1}$ decade$^{-1}$] (Eq. 3), and **g** and **h** atmospheric contribution to the mixed layer temperature tendency [Ks$^{-1}$ decade$^{-1}$] (Eq. 3).

**Historical (HIST) and North Atlantic-Global Atmosphere (NAGA) experiments.** We used the sixth version of the Model for Interdisciplinary Research on Climate (MIROC6)[29]. The horizontal resolution of the atmospheric component is a T85 spectral truncation (approximately 1.4° grid interval). There are 81 vertical levels, and the model top is set at 0.004 hPa. The ocean component is based on a tripolar coordinate system. The longitudinal grid spacing is 1°, and the meridional grid spacing varies from approximately 0.5° near the equator to 1° in the mid-latitudes. There are 62 vertical levels in the hybrid σ–z coordinate system.

After a 2000-year spin-up, a 800-year preindustrial control simulation was conducted[40]. 50 initial conditions were taken from the 800-year simulations with more than 10 years intervals. Using these initial conditions, historical simulation was conducted with the external forcing dataset following the protocol of CMIP6[30]. Historical simulations (HIST) were conducted with the 50 ensemble members from 1850–2014. In this study, only outputs after 1970 were analyzed. Because the historical simulations end in December 2014, SSP2-4.5 scenario simulations (i.e., a middle-of-the-road greenhouse gas emission scenario)[41] from 2015–2017 were combined with the historical simulation data. The SSP2-4.5 forcing was only used for the last three years, and thus we believe that the impact of the difference between the scenario selection is small in this study.

We also conducted North Atlantic-Global Atmosphere (NAGA) experiments using MIROC6, in which modeled SST anomalies are nudged to COBE-SST2 SST

anomalies at the time coefficient of three days over the Gulf Stream region (solid box; 30°–80° W and, 30°–50° N) (Supplementary Fig. 3b, d). Since the Gulf Stream SST warming in COBE-SST2 is smaller than HadISST2 (Fig. 1c), we expected that the usage of COBE-SST2 for SST nudging did not exaggerate the impact of the Gulf Stream warming. At the periphery of the restoration region from the solid box to the dashed box (24°–86° W and, 24°–56° N) (Supplementary Fig. 3b, d), the restored SST anomaly linearly decreases to 0 within 6 degrees. SST anomalies are defined as anomalies from climatology from 1970–2014 in the historical simulations (SST anomaly$_{model}$) and COBE-SST2 (SST anomaly$_{obs}$), respectively. For SST nudging, the nudging flux is added to the heat flux to the ocean surface:

$$Nudging\,flux = \frac{\rho C_p h}{\tau} \times \left( SST anomaly_{obs} - SST anomaly_{model} \right) \quad (1)$$

$\rho$ (=1027 kg m$^{-3}$) is the density of the seawater, $C_p$(=4187 J kg$^{-1}$ K$^{-1}$) is the specific heat of the seawater, $\tau$(=3 days) is the restoring time scale, and $h$ (=50 m) is assumed to be surface mixed layer affected by SST restoring. Several previous studies adapted this method (so-called pacemaker experiment) to detect the mechanism of climate change (e.g., global warming hiatus[42]). The reason for the application of the pacemaker experiment to the North Atlantic is that we expect that constraining SST in the Gulf Stream region improves the response of atmospheric or oceanic heat transport to the Barents-Kara Sea. The model is

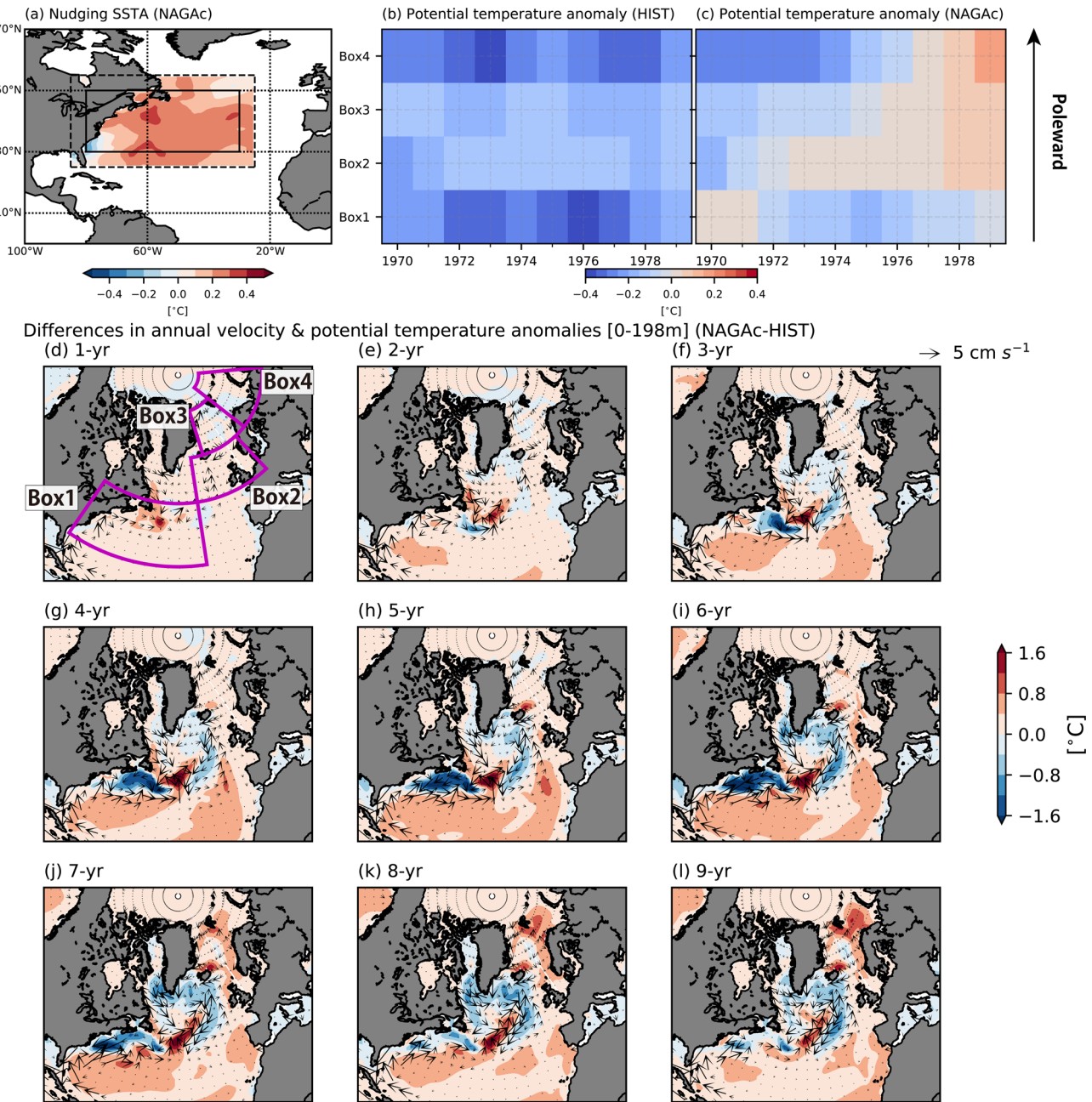

**Fig. 5 Time evolution of the potential temperature and horizontal velocity in the North Atlantic for NAGAc and HIST. a** SST anomaly pattern used for NAGAc experiments. The solid box shows the area in which SST anomalies are fully restored to the observed values. The restored SST anomalies are linearly reduced to 0 from the solid box to the dashed box as in NAGA (Methods). **b, c** Ensemble means of annual potential temperature anomalies for **b** HIST and **c** NAGAc area averaged in Gulf Stream region (Box1), Northeastern Atlantic (Box2), Greenland-Iceland-Norwegian Seas (Box3), and Barents-Kara Sea (Box4). The vertical axis indicates each box, while the horizontal axis indicates the year. The potential temperature is vertically averaged from the surface to 198 m depth. **d–l** Differences of the annual mean horizontal velocity [cm s$^{-1}$] (vector) and potential temperature [°C] (colors) between NAGAc and HIST from the first year to the ninth year (i.e., from 1970 to 1978). The potential temperature and horizontal velocity are vertically averaged from the surface to a depth of 198 m. Magenta solid boxes show areas used in **b** and **c**.

integrated from 1 January 1970 to 31 December 2017, and the initial conditions on 1 January 1970 are obtained from HIST. This experiment includes ten ensemble members corresponding to the initial conditions of each ensemble member in HIST, which are equivalent to CMIP6 historical runs (r1i1p1f1, r2i1p1f1… r10i1p1f1) by MIROC6. The external forcing data are the same as those in HIST; CMIP6 historical external forcing from 1970–2014 and SSP2-4.5 from 2015–2017.

To quantify the time evolution of ocean circulation response in the whole North Atlantic to the Gulf Stream SST warming, North Atlantic-Global Atmosphere with constant SST (NAGAc) experiments are also conducted. The settings used in NAGAc are similar to those in NAGA, but the modeled SST anomaly is restored to an idealized time-constant SST anomaly pattern. The linear trend pattern of COBE-SST2 over the Gulf Stream region for 1970-2017

(Fig. 5a; Supplementary Fig. 7) is used with the magnitude of which equivalent to a 10-year SST rise. The model is integrated from 1 January 1970 to 31 December 1979, and the initial conditions on 1 January 1970 are based on HIST with ten ensemble members.

**CMIP6 historical simulations**. Two sets of CMIP6 historical simulations for 1970-2014 are used in this study; one set is a multimodel ensemble based on 39 models (Supplementary Table 1). This multimodel ensemble consists of one member from each model. The other set is a multimodel ensemble of 13 CMIP6 models with more than 10 ensemble members (Supplementary Table 2). With this ensemble, we can estimate the external forced signal and the internal variability components in each model.

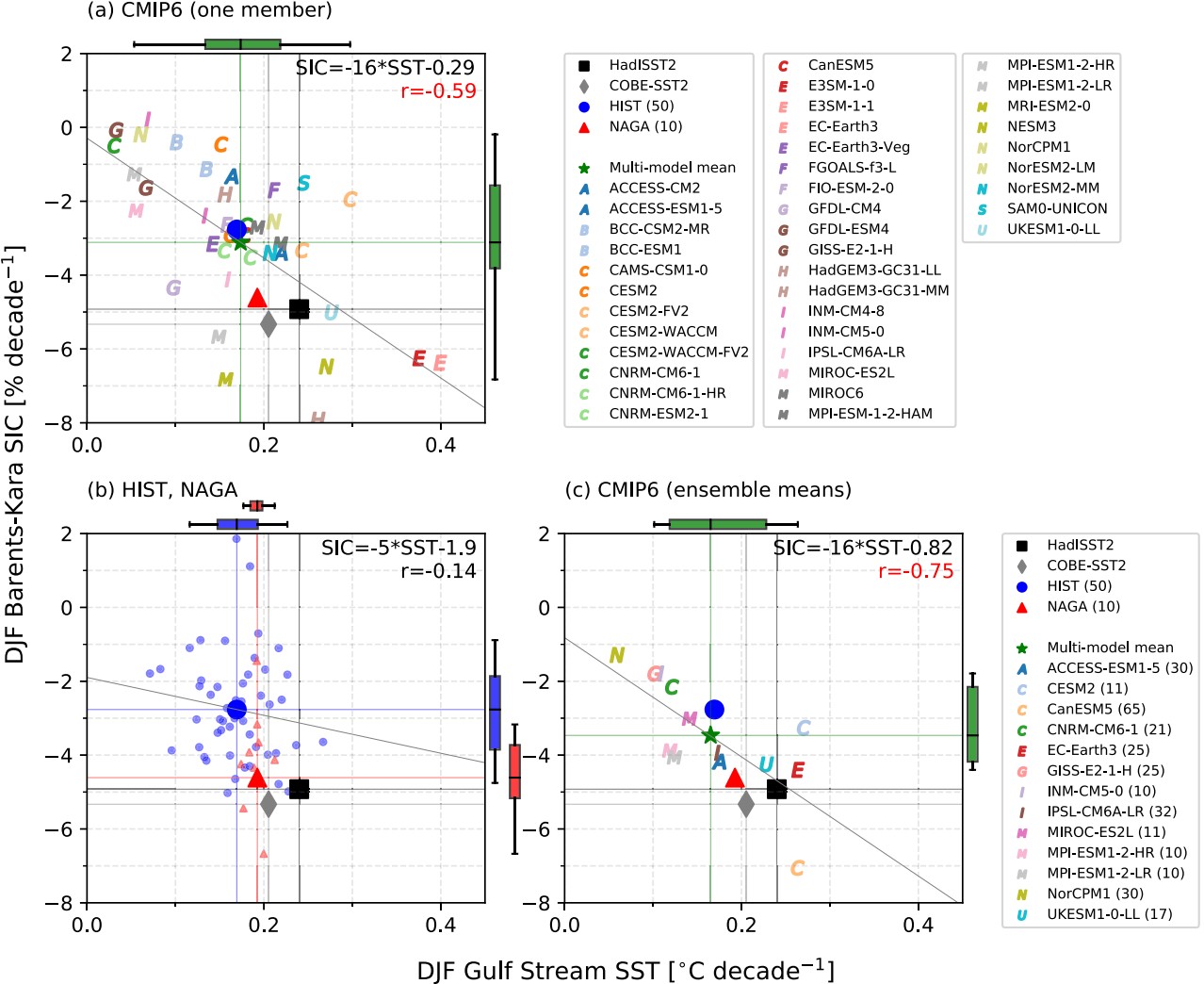

**Fig. 6 Relationships between the linear trends (1970–2014) in SIC and SST for HIST, NAGA, CMIP6, and observations. a** Scatter plots between the linear trends of DJF Gulf Stream SST [°C decade⁻¹] (horizontal axis) and Barents-Kara SIC [% decade⁻¹] (vertical axis) for one member of each CMIP6 model. Each model is denoted with different colored symbols, and the multimodel mean is indicated by a green star. The observations and ensemble means of HIST and NAGA are shown by black (HadISST2), gray (COBE-SST2), large blue (HIST ensemble mean), and large red (NAGA ensemble mean) markers. The least-squares linear fit for CMIP6 members is shown. The correlation coefficient and linear regression are shown in the panel. Vertical and horizontal lines indicate the SST and SIC trends for HadISST2 (black) and COBE-SST2 (gray), and multimodel means of CMIP6 (green). Box-and-whisker plots for SIC (right) and SST (top) trends are shown. The box extends from the 25% to 75% values of the data, with a line at the ensemble mean. The whiskers show the range from 5% to 95% of the data. **b** As in **a**, but for HIST and NAGA. Small blue markers indicate 50 ensemble members in HIST, and small red markers denote 10 members in NAGA. The least-squares linear fit for HIST members is shown. Vertical and horizontal lines indicate the SST and SIC trends for ensemble means of HIST (blue) and NAGA (red). **c** As in **a**, but for the ensemble means of CMIP6 models having more than ten ensemble members. Each model is denoted with different colored symbols, and the multimodel mean is indicated by a green star.

**Statistical analysis**. The linear trends of DJF SIC and SST in the Barents-Kara Sea and Gulf Stream regions are calculated by least-squares linear regression. The Barents-Kara SIC time series are defined as the area-averaged DJF mean SIC from 20°–70° E and, 65°–85° N. Gulf Stream SST time series are defined as the area-average of the DJF mean SST from 30°–80° W and, 30°–50° N. We also calculate the DJF SST averaged over the North Atlantic (0°–65° N, 0°–80° W) for the AMV index. To investigate the relationship between the Barents-Kara SIC trend and Gulf Stream SST trend in ensemble members, correlation coefficients between Barents-Kara SIC and Gulf Stream SST are calculated. Statistical tests for the correlation coefficients are performed based on a two-tailed Student's t-test at the 95% confidence level. The modified Mann-Kendall trend test for autocorrelated data is adapted for significance test of linear trends at 95% confidence level[43,44].

The arithmetic mean of all ensemble members is defined as the ensemble mean in this study. However, this definition may include the effect of the difference in the number of ensembles. Therefore, when we compare the differences in statistics for time series and linear trends (e.g., the ensemble mean, standard deviation, and quartiles) between HIST and NAGA, a comparison was made between the same ten ensemble members as follows. First, ten members are randomly selected from 50 members of HIST using the Monte Carlo method. Using selected ten members,

we calculate the ensemble mean and standard deviation in each year, and the ensemble mean and quartiles of linear trends for 1970–2017. This calculation is repeated 1000 times to obtain the average statistics. Time series and box-whisker plots shown in Fig. 1 represent the average and range of "statistics from ten members" for HIST and these are compared to single "statistics from ten members" for NAGA. Note that this Monte Carlo method is used to calculate the time series and the box-plot in Fig. 1. For the other analyses, only 10 HIST members that correspond to NAGA are used.

**Ratio of the ensemble mean variance to all variances**. The ratio of ensemble-mean variance to the total variance in the ensemble for X is calculated as below:

$$variance\ ratio = \frac{1}{n}\sum_{i}^{n}(X_{enmean,i} - \overline{X_{enmean}})^2 \bigg/ \frac{1}{N}\sum_{j}^{10}\sum_{i}^{n}(X_{j,i} - \bar{X}_j)^2 \qquad (2)$$

Here, $n$ is the length of the time, $X_i$ is $X$ for $t = i$, $X_{enmean}$ is the ensemble mean, $\bar{X}$ is the time mean of $X$, $X_{j,i}$ is $X$ for ensemble member $j$ and $t=i$, and $N$ (=$n \times 10$) is the total time length for 10 members. We refer to the numerator (denominator) of Eq. (2) as ensemble-mean variance (total variance in all ensembles). This is

analogous to the signal-to-noise ratio. When the variance ratio is 1, variations for all members are fully constrained to the ensemble mean.

**Ocean mixed layer heat budget analysis**. To diagnose heat balance in the upper ocean, we consider the temperature balance within the ocean mixed layer[45,46]:

$$\frac{\partial T_m}{\partial t} = \frac{Q_{net} - q_d}{\rho C_p H} - OCN \qquad (3)$$

$T_m$ means temperature averaged over the mixed layer, respectively. The first term on the right-hand side is the contribution from the surface heat flux (thus atmosphere), where $Q_{net}$ is the net surface heat flux, $q_d$ is the downward solar insolation penetrating through the bottom of the mixed layer, $\rho$ ($=1027 \, \mathrm{kg \, m^{-3}}$) is the density of the seawater, $C_p$($=4187 \, \mathrm{J \, kg^{-1} \, K^{-1}}$) is the specific heat of the seawater, and $H$ is the mixed layer depth. Here, we define $H$ as a depth where the density is $0.125 \, \mathrm{kg \, m^{-3}}$ higher than the surface density. We refer to the surface heat flux term as atmospheric contribution. The second term indicates oceanic contribution which includes the horizontal advection and the entrainment through the bottom boundary of the mixed layer. In this study, the oceanic term is calculated as the residual between the temperature tendency and atmospheric contribution.

**Decomposition of the horizontal subsurface heat flux**. The horizontal subsurface heat flux trends at 54 m depth are linearly decomposed into the contributions of temperature, velocity, and covariability between temperature and velocity; this relation is expressed as:

$$(\mathbf{u}t)_{\mathrm{trend}} = \mathbf{u}_{\mathrm{clim}}(t')_{\mathrm{trend}} + (\mathbf{u}')_{\mathrm{trend}}t_{clim} + (\mathbf{u}'t')_{\mathrm{trend}} \qquad (4)$$

where $\mathbf{u}$ and $t$ indicate the horizontal velocity and potential temperature, respectively. $()_{\mathrm{clim}}$ represents the 1970–2014 climatology for each detrended variable. $()_{\mathrm{trend}}$ is the linear trend for 1970–2017 and the prime means anomaly from the climatology for each variable.

**EOF analysis**. To extract the intermodel variations in the linear trends of winter SST in the Gulf Stream region, we performed EOF analysis for the linear trends of the DJF mean SST over 30°–80° W and 30°–50° N for CMIP6 climate models in two ways. First, an EOF was applied to the multimodel ensemble using 39 models, with a single member from each model considered. Second, a multimodel ensemble of 13 CMIP6 models with more than ten ensemble members (297 members in total) was considered. EOF analysis was also performed for the ensemble means of 13 models to extract intermodel variations in externally forced SST trends. Similarly, to extract the dominant mode of internal variability, EOF analysis was applied to deviations from the ensemble means of 297 members.

## Data availability

The HadISST2 dataset was downloaded from the Med Office website (https://www.metoffice.gov.uk/hadobs/hadisst2/data/download.html). The COBE-SST2 dataset was downloaded from the NOAA Physical Sciences Laboratory website (https://psl.noaa.gov/data/gridded/data.cobe2.html). The CMIP6 historical experimental dataset was downloaded from the ESGF website (https://esgf-node.llnl.gov/projects/cmip6/). Historical simulations of CMIP6 models were used in this study (Supplementary Table 1 and 2). The NAGA data for Figs. 1–6 generated in this study have been deposited in the Zenodo database[47] under https://doi.org/10.5281/zenodo.6445460.

## Code availability

Python scripts to reproduce the main figures have been deposited in the Zenodo database[47] under https://doi.org/10.5281/zenodo.6445460. The model code is available under restricted access for the developers' policy, access can be obtained by contact with the corresponding author upon reasonable request.

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

## Acknowledgements

This work is supported by the Ministry of Education, Culture, Sports, Science and Technology (MEXT), Japan, through the Integrated Research Program for Advancing Climate Models (TOUGOU, Grant Number JPMXD0717935457) (Y.Y., M.W., M.M., J.O.), the Program for the advanced studies of climate change projection (SENTAN, Grant Number JPMXD0722680395) (Y.Y., M.W., M.M., J.O.), the Arctic Challenge for Sustainability (ArCS) Project II (JPMXD1420318865) (M.M., J.O.), and JSPS KAKENHI Grant Number JP19H05703 (M.M.), JP20H05729 (Y.Y.), JP22H01299 (M.M.), JP22H04487 (Y.Y.), and JP22K14098 (Y.Y.). The model simulations were performed using Earth Simulator at the Japan Agency for Marine-Earth Science and Technology, Japan.

## Author contributions

Y.Y. and M.W. designed the research. Y.Y. performed the numerical experiments and analyses and wrote the paper. M.W., M.M., and J.O. helped with the analyses and the writing of the manuscript.

## Competing interests

The authors declare no competing interests.
