## [Peer Review File · Nature Communications]

Barents-Kara sea-ice decline attributed to surface warming in the Gulf StreamReviewers' Comments:

Reviewer #1:

Remarks to the Author:

Please see attached file.

Review of

'Barents-Kara sea-ice decline attributed to surface warming in the Gulf Stream'

By Y. Yamagami *et al.*

Submitted to *Nature Communications*

1. Key results

This study uses data from the global climate model MIROC6 (both CMIP6 [Coupled Model Intercomparison Project 6] simulations and nudged sea-surface temperature [SST] experiments) and CMIP6 models (historical simulations). The aim is to shed light on the relationship between SST in the Gulf Stream region and the mean sea-ice concentration (SIC) in the Barents-Kara Sea. The authors show that 56% of the negative SIC trend over 1970-2014 is explained by the externally-forced component of the Gulf Stream SST in CMIP6 models. They also find that when SST in the Gulf Stream region is nudged to observations in MIROC6, the modeled SIC trend is closer to observations, confirming that a strong link exists between Gulf Stream SST and Barents-Kara SIC. They further suggest a dynamical ocean link between the two via ocean circulation and ocean heat transport.

2. Significance

Understanding the causes of the recent Arctic sea-ice loss is extremely important in order to better understand our climate. Especially, the Barents-Kara Sea is an Arctic region where changes have been the strongest in winter over the past years. While many studies have recently focused on the role of ocean heat transport in driving sea-ice loss in the Barents Sea, not many analyses, to the best of my knowledge, have been carried out with more indirect causes, such as the warming in the Gulf Stream region. Thus, I think this study is highly significant in terms of advancing our knowledge in this field and further work in that direction would be highly recommended.

3. Major comments

The first main comment I have regarding this manuscript is that the hypothesis of the authors that Gulf Stream SST influences Barents-Kara SIC via ocean circulation is not really backed up by evidence. This would need to be solved to make this study really significant to the field. The main argument, if I understood correctly, is that the increased SST in the Gulf Stream region propagates in the northeast direction (Fig. 3), leading to enhanced ocean heat transport (Fig. 2), and resulting in decreased SIC in the Barents-Kara Sea (Figs. 1-2). While this argument makes sense, I am not convinced that the results show this dynamical link (see my specific comments in '4. Evidence'). This means that other ocean variables (e.g. transects of ocean temperature and velocity at depths between the Gulf Stream region and Barents-Kara Sea) would need to be analyzed to confirm this link. Also, maybe the atmospheric pathway / heat transport, which is mentioned in the Introduction (L51-54), is part of the dynamical link.

Second, I think the organization of the study could be improved by following the summary I provide in Section 1 ('Key results'). To me, the first key result of the study is that there is a clear anti-correlation between the Barents-Kara SIC trend and the externally-forced Gulf Stream SST trend, as shown by CMIP6 models (what you discuss in 'SIC decrease and Gulf Stream warming in CMIP6 climate models'). Then, I think the second main result is that nudging Gulf Stream SST to observations leads to improved SIC trend in Barents-Kara Sea (what you discuss in 'Ensemble NAGA experiment'). Finally, trying to find the dynamical link between Gulf Stream SST and Barents-Kara SIC would be the third key result (what you discuss in 'Processes responsible for the Barents-Kara sea-ice decline' and 'Ocean circulation change'). For this third result, more evidence would be needed, as emphasized in my previous main comment.

A last main comment is that in some parts (especially the Methods), the paper lacks clarity that would help in better understanding what the authors did. I think it is important that the authors solve these issues. Please see my specific comments below.

Below I provide specific comments in order to help the authors to improve their manuscript, and I divide them into the following sections: '4. Evidence', '5. Validity', '6. Data and methodology', '7. Clarity and context', '8. References'.

4. Evidence

L103-125: Results from this section are nice (and logical). But I don't really understand (at this stage at least) the link that you want to establish by analyzing trends in horizontal heat flux and potential temperature in the Barents-Kara Sea. In the end, you want to prove that the Gulf Stream SST influences the Barents-Kara SIC. You need to provide a justification why you look at these specific variables in the beginning of this section (e.g. Gulf Stream SST can influence ocean heat transport to the Barents Sea, which influences sea ice there). By the way, at L107, you write 'surface ocean heat fluxes' but this is the 54m deep horizontal heat flux, so you need to correct it (e.g. 'subsurface ocean heat fluxes'). Finally, in this section, I think you really need to highlight that your key result is the increase in ocean heat transport to the Barents Sea in NAGA compared to HIST, which supports the hypothesis that the increase in the Gulf Stream SST (coming from the nudging) leads to enhanced ocean heat transport, resulting in decreased Barents-Kara SIC.

L138-140 and L402-406: I don't really understand what you did in the NAGAc experiments. What do you mean by 'constant SST magnitude'? Is it constant in space or in time? Please better explain the purpose and the method of these experiments. Additionally, I don't fully agree that Fig. 3c-k demonstrates that the Gulf Stream warming reaches the Barents-Kara Sea after 7 years. First, in NAGAc, it is only the northeastern part of the Gulf Stream region that warms compared to HIST, and there is a relatively strong cooling in the western part. Second, you don't really see that this warming spot 'propagates' to the northeast towards the Barents Sea, it rather stays in place. Third, it is true that the Norwegian Sea / Barents Sea starts to warm after 7 years, but since there is a cooling spot between the Barents Sea and the Gulf Stream (which you mention at L134-136), how are you sure that this is a result of the Gulf Stream warming propagation? Shouldn't you show results after more than 9 years in Fig. 3?

L202-204: While you show a robust connection between trends in Gulf Stream SST and Barents-Kara SIC (Fig. 4b, Supplementary Fig. 8), I am not totally convinced by the dynamical link you suggest through ocean circulation. Figure 3 shows a region of warming between NAGAc and HIST in the northeast of the Gulf Stream region, a cooling in the region south of Iceland and a slight warming in the Norwegian / Barents Sea, but I can't see evidence of a northeastward propagation of the Gulf Stream warming via ocean circulation (see also my previous comment L138-140). In order to make such a conclusion, I think you would really need to come up with more evidence. Maybe the atmospheric pathway, which you don't investigate here (but mention in your introduction), is also important and drives part of sea-ice loss in the Barents Sea?

5. Validity

L20-21 and L36-37: You write that 'the past sea-ice decrease rate is underestimated in the majority of CMIP6 models'. But I think this statement is not totally accurate as it depends on the time period (1979-2014 or another period?), the season (March or September?), the models considered (all available models or a subset?) and the region (total Arctic or Barents-Kara Sea?). It might be true for Barents-Kara Sea (as suggested by your Supplementary Fig. 2c,d), but it is not the case for the total Arctic (north of 40°N) based on a personal analysis of all available CMIP6 models. In your text, it is not clear you are talking about the total Arctic or the Barents-Kara Sea when you mention this in L20-21 and L36-37, but I suspect this is for the total Arctic as you make reference to Stroeve et al. (2012, <https://doi.org/10.1029/2012GL052676>), who have investigated CMIP3 and CMIP5 models (and not CMIP6). Over 1979-2014 (1979 being the beginning of satellite observations and 2014 being the end of the CMIP6 historical simulations), the trend in March Arctic sea-ice area is more negative (so larger decrease rate) than observed in 28 CMIP6 models over a total of 49 models (so 57% of the models), and the multi-model mean (-380,000 km² per decade) also has a more negative trend in March sea-ice area than observed (-290,000 km² per decade). For September, a lower number of models shows a more negative trend in Arctic sea-ice area than observed over 1979-2014, but there are still 16 models (so 33%), and the multi-model mean (-670,000 km² per decade) has a less negative trend in September sea-ice area than observed (-800,000 km² per decade). If you look at Figure 2 of SIMIP Community (2020, <https://doi.org/10.1029/2019GL086749>), which you mention in your text, you will see that the multi-model mean trend in March Arctic sea-ice area is more negative than observed and the multi-model mean trend in September Arctic sea-ice area is relatively close to observations, except over the very last years (from around 2007, when there is an acceleration of the observed decrease in sea-ice area). As this statement partly motivates your study, I strongly recommend to re-adjust your text and be much more precise than currently by clearly specifying the period, the season, the models considered and the exact region for this specific statement. You also need to make reference to Supplementary Fig. 2c,d, which I suggest to merge with Fig. 1c,d as this is an important result (you could add CMIP6 results in Fig. 1). For your study, I think it makes more sense to focus only on Barents-Kara Sea and over the winter season for this statement (so no need to mention the total Arctic and summer time).

L58-59: Related to my previous comment, you mention an 'underestimation of the long-term decrease in sea-ice concentration'. What do you mean exactly? Do you mean that the trend in CMIP6 multi-model mean March Barents-Kara SIC is less negative than observed? If yes, please be more precise and relate it to your Supplementary Fig. 2c,d,

which I suggest to merge with Fig. 1c,d as this is an important result. Also, what is SIC here: is it the mean SIC over the entire Barents-Kara region? You define it in the Methods but it would also be good to briefly define it here.

L85-89: I agree that the SIC trend becomes much closer to observations with NAGA compared to HIST (Fig. 1c,d). However, for SST, as you nudge to COBE-SST2, the NAGA trend logically becomes closer to COBE-SST2 compared to HIST, but it is still $0.05^{\circ}\text{C}/\text{decade}$ off from HadISST2 (Fig. 1a,b). This is due to a relatively large difference in SST trend between the two observational datasets (0.21 vs. $0.24^{\circ}\text{C}/\text{decade}$). Why did you choose COBE-SST2 for the nudging and not HadISST2? You should provide a reasoning for this? Otherwise, one would ask what would happen if you nudged to HadISST2 instead, which has a larger trend in SST. Related to that, you can see that there is a large improvement in SIC trend when using NAGA instead of HIST, relative to observations (which are relatively close to each other). However, the improvement is not that clear for SST trend as the difference between the two observational datasets ($0.03^{\circ}\text{C}/\text{decade}$) is comparable to the difference between NAGA and HIST ($0.02^{\circ}\text{C}/\text{decade}$). I think you should comment on this.

6. Data and methodology

L75 and L398-399: Did you start the 10 members of NAGA from 10 different MIROC6 CMIP6 members? If yes, how did you select them? It is not clear from the text.

L121: According to Supplementary Fig. 6, this is the horizontal SURFACE heat flux. You should write it. You should also add this precision to the Methods (L440-448).

L387: A short description on how the 50 MIROC6 members are generated is necessary, as this is the main model used in this study.

L388: It should be 'SSP2-4.5' instead of 'SSP245'. Why do you use this specific scenario and not another one? A small justification is needed.

L390-393: As this is an important method used in your study, you should describe how you nudged SST anomalies to observations. Did you use a correction in the surface heat flux or something else? Do you only nudge at the beginning of the simulation or do you update it through time? Additionally, you need to provide a motivation why you perform this set of experiments with a sentence or two.

L402-406: What do you exactly mean by 'constant SST with the magnitude equivalent to a decade of temperature rise from 1970-2017 in COBE-SST2'? What is the purpose of this experiment?

L430-439: I understand the logic to randomly select 10 members in HIST to correctly compare to the 10 members of NAGA. But if I understood correctly, the 10 NAGA members were started from 10 different HIST members, is it right? (see my comment L75) If yes, then shouldn't you simply take the 10 HIST members that correspond to NAGA instead of making a random selection?

L461: A description of observational datasets (COBE-SST2 and HadISST2) is missing in the Methods. Also I didn't find a definition of the acronym COBE-SST2 in your text.

L615-618: EC-Earth3 also has more than 10 members for the historical CMIP6 simulations. Why didn't you use this model?

7. Clarity and context

L33-36: This sentence is confusing as it mixes both future model projections of Arctic sea ice (in the first part of the sentence) and recent observations (in the second part of the sentence). I think it would be better to state that 'there is observational and modeling evidence that the retreat of Arctic sea ice has been driven by anthropogenic greenhouse gas emissions'. Also, the references you provide are a bit outdated and there have been more recent studies related to this, e.g. Notz & Stroeve (2016, <https://doi.org/10.1126/science.aag2345>) and Stroeve & Notz (2018, <https://doi.org/10.1088/1748-9326/aade56>).

L46 and other instances in the text: You always write 'Barents-Kara sea-ice variability', but this is not precise. You should rather say 'Barents-Kara sea-ice area (or extent or concentration) variability'.

L48: 'the winter sea-ice tends to be less...': less what? Don't you mean that the winter sea-ice concentration or area tends to be lower?

L62-63: As it is, the reader can have the feeling that it is the study of Kosaka & Xie (2013) that demonstrated that the pacemaker experiment provides better results than CMIP6 historical runs. I suggest you remove the reference '30' to Kosaka & Xie (2013), which is a very different study, and re-write: 'Our pacemaker experiment demonstrates that the Barents-Kara...'. Also, it is not clear what you mean by SIC: is it the mean SIC over the whole Barents-Kara Sea? Wouldn't it be better to use sea-ice area or extent instead?

L63: You miss something after 'can be better reproduced'. I guess you mean that the SIC trend can be better reproduced if SST anomalies in the Gulf Stream region are constrained by observations.

L69: I wouldn't use an acronym (NAGA) in the section title, but rather the full name for clarity.

L71: You need to add that MIROC6 is a global climate model, especially for the audience of Nature Communications.

L73: 'The other set of experiments is...'

L74: What do you mean by 'strongly restored'? Is 'strongly' necessary?

L77: Please write the months taken into account for the winter average: is it DJF?

L87: In fact, the NAGA trend is lower than HIST (as there is a minus sign)... You should instead write 'more negative' or 'larger in absolute value'.

L92: As this metric is important, you need to better describe what is the 'ensemble mean variance' and what is the 'total variance in the ensemble'.

L105-106: What are 'linear trends of oceanic states'?

L130-132: Please be more precise: 'In NAGA, both the velocity and temperature trends (vertically-averaged over the first 200 m) in the Gulf Stream region and Norwegian Sea are larger than those in HIST. Also, the warming in the Gulf Stream...'

L133-134: The Gulf Stream slowdown you mention is confusing as you previously talk about larger temperature and velocity in the Gulf Stream region in NAGA (L130-131). I think you want to say that there is a small spot of cooling at about 40°W / 45°N, and also the width of the Gulf Stream in the eastern branch seems to weaken in NAGA compared to HIST. Is that correct? If yes, please be more precise.

L151: I wouldn't use the SIC acronym (but rather the full name instead) in the section title for the clarity of the paper.

L161: You need to make reference to Supplementary Table 2.

L173: What do you mean by 'the variance of the SIC trend'?

L177: I don't clearly understand what you did when you say that you removed the multi-model mean trend.

L217-218: For Fig. 1a,c, I suggest to write SST and SIC 'anomalies', respectively, in the Y axis label.

L221: There are several lines in Fig. 1a,c, so please use the plural. Also, linear trends in Fig. 1a,c are not shown by dashed lines but rather by thick solid lines. But for the clarity of your figure, I would recommend using dashed lines, as you state in the caption.

L225: This is not the 'number of linear trends', this is rather the trend in SST / SIC for each dataset.

L244: This is not the surface temperature and velocity but rather vertically-averaged temperature and velocity in the first 198 m.

L393-394: You should be more explicit and make reference to Supplementary Fig. 3b as this is where the solid box is.

L408: Shouldn't it be 31 December 2017 instead of 1979?

L521-524: The two first sentences of the caption of Supplementary Fig. 3 are redundant. Please consider merging them together. Also, it is not clear what you mean by 'ensemble mean variance' and 'total ensemble member variance' (also present at L92). Which period is considered here?

L536: The unit '°C' is missing for 'potential temperature [°C decade⁻¹]'.

L553: In order to make this figure visually more attractive, I would color the map with the magnitude of surface heat flux. You can let the vectors showing the direction of surface heat fluxes.

L567: This is a nice figure but the legend text could be enlarged for clarity. Maybe you could include this figure as a panel of Fig. 4 to better support your conclusions.

8. References

L46-56: You mention two main drivers of Barents-Kara SIC at interannual time scales, namely ocean heat transport and Gulf Stream SST meridional shift. In your study, you make the hypothesis that it is in fact a combination of the two that is responsible for the sea-ice loss. However, in this part, you only say that the Gulf Stream meridional shift could enhance the atmospheric heat transport and you provide references for this. What about references that make a connection between Gulf Stream SST and ocean heat transport? If they exist, they should be cited here. If not, I think it should also be said.

L71: You need to add a reference for MIROC6, even if done in the Methods.

L73: You need to add a reference to the CMIP6 description paper (Eyring et al., 2016, <https://doi.org/10.5194/gmd-9-1937-2016>).

L109: As this is your result, I wouldn't provide a reference to Arthun et al. (2012) here.

L283: You should revise the reference to SIMIP Community: it should be 'SIMIP Community (2020)...' and not 'Community, S. I. M. I. P. (2020)...'

Reviewer #2:

Remarks to the Author:

Review of NCOMMS-21-24661-T

"Barents-Kara sea-ice decline attributed to surface warming in the Gulf Stream" by Yamagami, Y., Watanabe, M., Mori, M. & Ono, J.

This study presented the impact of Gulf Stream warming on sea ice decline over the Barents-Kara Seas in winter using ensemble climate models. Authors found that there is a negative relationship between linear trends of upper layer ocean temperature over the Gulf Stream and sea ice concentration over the Barents-Kara Seas from 1970 to 2017. The Gulf Stream warming slowly propagates northeastward along the European region and reaches the Barents-Kara sector, inducing sea ice decline over the Barents-Kara Seas. To my knowledge, this is a new study to show link between Atlantic ocean variability and Arctic ocean climate. It will be a good piece of work for better understanding Northern Hemisphere climate variability. However, there are a few things that could be done to strengthen this paper.

It is not enough for discussing the impact of atmospheric circulation variability over the Northern Hemisphere on sea ice decrease over the Barents-Kara Seas. Previous studies reported that the atmospheric response to warming over the Gulf Stream has direct and/or indirect impact on Arctic sea ice decline. In addition, over the Barents-Kara sector, northward movement of sea ice is associated with anomalous southerly winds reported by previous studies. I think authors should discuss results of linear trends of atmospheric circulations.

From difference in upper-level ocean temperature between NAGAc and HIST, the Barents-Kara Seas warming is clear after 7 years, indicating warm ocean temperature anomaly reaches Barents-Kara Seas sectors. However, from the annual mean horizontal velocity in figure 3, it is unclear to me that warming temperature anomaly propagates northeastward along the European coast after 3 years later, in particular in 4 and 5 years. I recommend authors show northward advection of warm temperature anomaly from subtropical gyre region to the European coast region clearly.

Therefore, I suggest reconsideration of this paper after major revisions.

Specific comments:

1) To discuss the impact of surface atmospheric parameters on sea ice concentrations, authors should show linear trends of atmospheric parameters (e.g. sea level pressure, temperature, and wind speed in the lower troposphere). The geopotential height at upper troposphere (e.g. 300hPa) would be good for discussing the teleconnection between mid- to high-latitudes.

2) If we focus on warming ocean temperature anomaly during 7 years in NAGAc, this warming anomaly seems to move northeastward from the subtropical gyre region to the Norwegian Sea via the European coast. However, to my knowledge, the northward movement of ocean current is weak over the subtropical gyre, suggesting that ocean current would have a small impact on the propagation of warm temperature anomaly. The northward movement of ocean current from the subtropical gyre to the European coast region is clearly seen in the climate model? or there is another mechanism for this propagation without ocean current?

3) I am not sure the Gulf Stream warming anomaly reaches the Barents-Kara sea ice decreasing trend area (eastward of 50E). To show advection of warming anomaly to sea ice decline area clearly, I would like to see the extended areas over the Barents-Kara sea (e.g. extended to 60E).

4) The horizontal heat flux and ocean temperature at a depth of 54m were shown in Figure 2. However, the sea ice is influenced by ocean temperature and current near-surface. Therefore, authors should show averaged horizontal heat flux and ocean temperature from the surface to a depth of several tens or several hundred such as figure 3, or show vertical distributions of potential

temperature with horizontal heat flux over sea ice decreasing trend area such as Supplementary Figure 4a-c (e.g. 50E, 70N to 85N).

5) Author conducted statistical analysis for figures 2 and 3? I think it is important that upper-level ocean temperature in NAGAc has statistically significant warming. Please show areas with statistically significant trends.

Reviewer #1:

Understanding the causes of the recent Arctic sea-ice loss is extremely important in order to better understand our climate. Especially, the Barents-Kara Sea is an Arctic region where changes have been the strongest in winter over the past years. While many studies have recently focused on the role of ocean heat transport in driving sea-ice loss in the Barents Sea, not many analyses, to the best of my knowledge, have been carried out with more indirect causes, such as the warming in the Gulf Stream region. Thus, I think this study is highly significant in terms of advancing our knowledge in this field and further work in that direction would be highly recommended.

- We would like to thank the reviewer for your time and effort to read our manuscript. The reviewer's constructive comments encouraged us to conduct additional analyses and substantially revised the manuscript. We hope that the revised manuscript will address all reviewer's concerns.

The first main comment I have regarding this manuscript is that the hypothesis of the authors that Gulf Stream SST influences Barents-Kara SIC via ocean circulation is not really backed up by evidence. This would need to be solved to make this study really significant to the field. The main argument, if I understood correctly, is that the increased SST in the Gulf Stream region propagates in the northeast direction (Fig. 3), leading to enhanced ocean heat transport (Fig. 2), and resulting in decreased SIC in the Barents- Kara Sea (Figs. 1-2). While this argument makes sense, I am not convinced that the results show this dynamical link (see my specific comments in '4. Evidence'). This means that other ocean variables (e.g. transects of ocean temperature and velocity at depths between the Gulf Stream region and Barents-Kara Sea) would need to be analyzed to confirm this link. Also, maybe the atmospheric pathway / heat transport, which is mentioned in the Introduction (L51-54), is part of the dynamical link.

- We thank the reviewer for your critical and constructive comments on the dynamical link between Gulf Stream SST and Barents-Kara SIC trends. To address the reviewer's concerns, we have conducted two additional analyses.
- First, to highlight the role of ocean heat transport on the Barents-Kara SIC reduction against the atmospheric influence, we have investigated linear trends of lower atmospheric circulation and temperature (Fig. 2c, d). This analysis has revealed the minor role of the atmospheric pathway from the Gulf Stream to the Barents-Kara Sea. To explain this result,

we have also added the following sentences in the revised manuscript (Line 112-129):

- First, the linear trends of surface ocean and lower atmosphere are compared between HIST and NAGA over the Barents-Kara Sea to determine whether atmospheric or oceanic variability contributes to the decrease in SIC (Fig. 2). Differences in surface wind trends appear to cause sea-ice to retreat more poleward in NAGA, but the trends are not statistically significant (Fig. 2c,d). Moreover, sea-ice drift trends in NAGA and HIST respond to surface ocean circulation changes rather than wind changes (Fig. 2a-f), suggesting the differences in the dynamical contribution of the atmosphere to sea-ice loss are not the cause of the sea-ice loss difference between NAGA and HIST. While, the spatial agreement of negative trends in NAGA between the SIC, surface salinity, and surface heat fluxes (Supplementary Fig. 4a,c) means that the atmospheric surface warming trend over that region (Fig. 2c) is the result of sea-ice loss rather than the cause. The same is true in the difference between NAGA and HIST. The more considerable SIC decrease in NAGA is not due to the difference in atmospheric heat advection, which is different in the case of interannual variability²⁶. The heat release difference is due to the less sea-ice formation (i.e., the less SIC), which is implied by the more negative trends of surface salinity flux in NAGA (Supplementary Fig. 4c,d). Therefore, the improved SIC trend in the Barents-Kara Sea results from the SST warming (Fig. 2e, f), possibly driven by oceanic heat transport from the North Atlantic domain.
- Second, based on the reviewers' comments and suggestions, we have examined the transects of ocean potential temperature and horizontal heat transport to the north of 65°N, to show the dynamical link between the Gulf Stream and the Barents-Kara Sea (Fig. 3). It is found that the poleward heat transport from the North Atlantic domain increases more in NAGA than HIST, which leads to the faster reduction of Barents-Kara SIC in NAGA. To explain this result, we have also added the following paragraph in the revised manuscript (Line 130-138):
 - To show the dynamical link between the North Atlantic and the Barents-Kara Sea, we examine the transects of ocean potential temperature and horizontal heat transport to the north of 65°N (Fig. 3). The large difference is found in the northern part of the Barents-Sea Opening section at 20°E. The heat transport increase in the northern section at 20°E is different from the observed results¹⁸ possibly due to the coarse resolution of MIROC6. The trends of poleward surface-subsurface heat transport in the Norwegian Sea (70°N) and south of Iceland (65°N) are weakly negative in HIST, but positive in NAGA (Fig. 3f,i). This increase of the poleward ocean heat transports from the North Atlantic is considered to warm the Barents-Kara Sea in NAGA more than HIST.

Second, I think the organization of the study could be improved by following the summary I provide in Section 1 ('Key results'). To me, the first key result of the study is that there is a clear anti-correlation between the Barents-Kara SIC trend and the externally-forced Gulf Stream SST trend, as shown by CMIP6 models (what you discuss in 'SIC decrease and Gulf Stream warming in CMIP6 climate models'). Then, I think the second main result is that nudging Gulf Stream SST to observations leads to improved SIC trend in Barents- Kara Sea (what you discuss in 'Ensemble NAGA experiment'). Finally, trying to find the dynamical link between Gulf Stream SST and Barents-Kara SIC would be the third key result (what you discuss in 'Processes responsible for the Barents-Kara sea-ice decline' and 'Ocean circulation change'). For this third result, more evidence would be needed, as emphasized in my previous main comment.

- We thank the reviewer for your constructive suggestion on the manuscript's organization. We have newly added SST and SIC trends in the CMIP6 models in Fig. 1 (Fig. 1b, d), based on your comment.
- On the other hand, we have decided to discuss the correlation between SIC and SST in the CMIP6 model after "Processes responsible for the Barents-Kara sea-ice concentration decline" and "Ocean circulation change". This is because the motivation for examining the correlation between SST and SIC in CMIP6 multi-model has risen only after we find the dynamical link between the SST and SIC trends in NAGA and HIST. Therefore, we first investigate the mechanism of changes in the NAGA experiment, and then we show the results for the CMIP6 models.
- For the third result pointed out by the reviewer (i.e., what we discuss in "Processes responsible for the Barents-Kara sea-ice concentration decline" and "Ocean circulation change"), we have conducted additional analyses which support the dynamical link between SST and SIC, as mentioned in response to the first major comment.

A last main comment is that in some parts (especially the Methods), the paper lacks clarity that would help in better understanding what the authors did. I think it is important that the authors solve these issues. Please see my specific comments below. Below I provide specific comments in order to help the authors to improve their manuscript, and I divide them into the following sections: '4. Evidence', '5. Validity', '6. Data and methodology', '7. Clarity and context', '8. References'.

- We appreciate the reviewer's specific comments. We have thoroughly revised the manuscript

following each comment to improve its clarity. The specific responses are given below.

L103-125: Results from this section are nice (and logical). But I don't really understand (at this stage at least) the link that you want to establish by analyzing trends in horizontal heat flux and potential temperature in the Barents-Kara Sea. In the end, you want to prove that the Gulf Stream SST influences the Barents-Kara SIC. You need to provide a justification why you look at these specific variables in the beginning of this section (e.g. Gulf Stream # SST can influence ocean heat transport to the Barents Sea, which influences sea ice there).

- Following the reviewer's suggestion, to explain why we examine particular variables, we have added the following sentences to the beginning of this paragraph (Line 109-114):
 - Given the importance of links between the North Atlantic SST and the Barents-Kara SIC variation as suggested in the literature²⁵, the further upstream Gulf Stream SST can be the source of the ocean and/or atmospheric heat transport to the Barents Sea, which influence sea-ice decline there. First, the linear trends of surface ocean and lower atmosphere are compared between HIST and NAGA over the Barents-Kara Sea to determine whether atmospheric or oceanic variability contributes to the decrease in SIC (Fig. 2).
- We note that the new analysis shown in this paragraph suggests that ocean heat transport from the North Atlantic domain is essential rather than atmospheric influence, as we mentioned in reply to the first main comment.

By the way, at L107, you write 'surface ocean heat fluxes' but this is the 54m deep horizontal heat flux, so you need to correct it (e.g. 'subsurface ocean heat fluxes').

- Although we agree with the reviewer, we have removed these sentences about subsurface ocean heat fluxes since we have changed the variables to analyze in this paragraph. In the third paragraph of this section, we have mentioned the horizontal heat flux at 54m depth as 'subsurface ocean heat fluxes' instead (Line 142 and 148).

Finally, in this section, I think you really need to highlight that your key result is the increase in ocean heat transport to the Barents Sea in NAGA compared to HIST, which supports the hypothesis that the increase in the Gulf Stream SST (coming from the nudging) leads to enhanced ocean heat transport, resulting in decreased Barents-Kara SIC.

- We thank the reviewer's constructive comment, which motivated us to examine the linear trends of ocean heat transport from the North Atlantic to the Barents-Kara Sea (Fig. 3), as we

mentioned in the reply to the first main comment. This additional analysis revealed that ocean heat transport in NAGA increases more than HIST from the Norwegian Sea to the Barents-Kara Sea. We have added the explanation on ocean heat transport in the Norwegian Sea to this section. Also, the heat transport change from the Gulf Stream to the Norwegian Sea is discussed in the next section (“Ocean circulation change”).

L138-140 and L402-406: I don’t really understand what you did in the NAGAc experiments. What do you mean by ‘constant SST magnitude’? Is it constant in space or in time? Please better explain the purpose and the method of these experiments. Additionally, I don’t fully agree that Fig. 3c-k demonstrates that the Gulf Stream warming reaches the Barents-Kara Sea after 7 years. First, in NAGAc, it is only the northeastern part of the Gulf Stream region that warms compared to HIST, and there is a relatively strong cooling in the western part. Second, you don’t really see that this warming spot ‘propagates’ to the northeast towards the Barents Sea, it rather stays in place. Third, it is true that the Norwegian Sea / Barents Sea starts to warm after 7 years, but since there is a cooling spot between the Barents Sea and the Gulf Stream (which you mention at L134-136), how are you sure that this is a result of the Gulf Stream warming propagation? Shouldn’t you show results after more than 9 years in Fig. 3?

- We deeply appreciate the reviewer’s critical comment. This motivates us to revise this section substantially. We have added the reason for implementing the NAGAc in the Methods and fixed the explanation of NAGAc settings that was difficult to understand. (Line 515-523). The NAGAc aims to investigate the time-evolution of surface ocean temperature and velocity anomalies in the whole North Atlantic to SST warming in the Gulf Stream. In this experiment, SST anomalies over the Gulf Stream region are restored to an idealized positive SST anomaly pattern in which spatial structure and magnitude are constant in time.
- To show the result in NAGAc more clearly, we have added the new figure (Fig. 5). The idealized SST anomaly is shown, whose amplitude is equivalent to the decadal SST increase in the 1970-2017 trend of COBE-SST2 (Fig. 5a). The result demonstrates that surface temperature warming in the Gulf Stream region gradually extends northeastward and makes a significant difference compared to HIST in the Barents-Kara Sea after about seven years (Fig. 5b-1).
- Also, we have conducted the mixed layer heat budget analysis to highlight the role of ocean dynamics on surface ocean warming. This additional analysis revealed that the oceanic contribution in NAGA is larger than HIST and extends to the downstream of the Gulf Stream, which corresponds to the larger temperature warming in the Norwegian Sea (Fig. 4e,f).

While, the cooling trend between the Gulf Stream and the Norwegian Sea is due to enhanced cooling by surface heat fluxes in the subpolar region (Fig. 4g,h), which does not mean a break in poleward ocean heat transport. To show these results, we have added the following paragraph (Line 164-173):

- In both HIST and NAGA, ocean dynamics contribute to surface temperature warming in the Norwegian Sea (Fig. 4e,f), but its contribution is larger in NAGA and extends to the downstream of the Gulf Stream (around 30°W and 50°N), which is consistent to the larger temperature warming in the Norwegian Sea (Fig. 4c,d). The positive ocean contribution trend is found along the acceleration trends of surface velocity, suggesting that poleward ocean heat transport strengthens (Fig. 4e, f). In NAGA, the positive temperature trend between the Gulf Stream and the Norwegian Sea is interrupted by a negative trend (Fig. 4c,d). This is due to enhanced cooling by surface heat fluxes in the subpolar region (Fig. 4g,h), which do not mean a break in poleward ocean heat transport.
- Since NAGAc investigates the time evolution of surface temperature and velocity in the North Atlantic against the idealized SST anomalies, SST anomalies in NAGAc do not reflect the realistic SST variations. Therefore, performing long-term integrations is not appropriate for NAGAc. In addition, in NAGAc, the constant heat input in the Gulf Stream region occurs, leading to too large poleward heat transport in the long-term integration.

L202-204: While you show a robust connection between trends in Gulf Stream SST and Barents-Kara SIC (Fig. 4b, Supplementary Fig. 8), I am not totally convinced by the dynamical link you suggest through ocean circulation. Figure 3 shows a region of warming between NAGAc and HIST in the northeast of the Gulf Stream region, a cooling in the region south of Iceland and a slight warming in the Nowegian / Barents Sea, but I can't see evidence of a northeastward propagation of the Gulf Stream warming via ocean circulation (see also my previous comment L138-140). In order to make such a conclusion, I think you would really need to come up with more evidence. Maybe the atmospheric pathway, which you don't investigate here (but mention in your introduction), is also important and drives part of sea-ice loss in the Barents Sea?

- As we mentioned in the first main comment and comment on L138-140 and L402-406, we have performed two additional analyses. First, we have investigated linear trends of lower atmospheric circulation and temperature (Fig. 2c, d). This analysis has revealed the minor role of the atmospheric pathway from the Gulf Stream to the Barents-Kara Sea. Second, we have conducted the mixed layer heat budget analysis and re-examined the ocean circulation response in the NAGAc. This additional analysis showed that surface ocean warming by

ocean dynamics is enhanced from the downstream of the Gulf Stream, which transports the additional heat to the Barents-Kara Sea.

- In both HIST and NAGA, the temperature warming trends in the Gulf Stream region and the Norwegian Sea appear to be broken by cooling trends between them. However, this is due to enhanced cooling by surface heat fluxes in the subpolar region (Fig. 4g, h), which does not mean a break in poleward ocean heat transport.

5. Validity

L20-21 and L36-37: You write that ‘the past sea-ice decrease rate is underestimated in the majority of CMIP6 models’. But I think this statement is not totally accurate as it depends on the time period (1979-2014 or another period?), the season (March or September?), the models considered (all available models or a subset?) and the region (total Arctic or Barents-Kara Sea?). It might be true for Barents-Kara Sea (as suggested by your Supplementary Fig. 2c,d), but it is not the case for the total Arctic (north of 40°N) based on a personal analysis of all available CMIP6 models. In your text, it is not clear you are talking about the total Arctic or the Barents-Kara Sea when you mention this in L20-21 and L36-37, but I suspect this is for the total Arctic as you make reference to Stroeve et al. (2012, <https://doi.org/10.1029/2012GL052676>), who have investigated CMIP3 and CMIP5 models (and not CMIP6). Over 1979-2014 (1979 being the beginning of satellite observations and 2014 being the end of the CMIP6 historical simulations), the trend in March Arctic sea-ice area is more negative (so larger decrease rate) than observed in 28 CMIP6 models over a total of 49 models (so 57% of the models), and the multi-model mean (-380,000 km² per decade) also has a more negative trend in March sea-ice area than observed (-290,000 km² per decade). For September, a lower number of models shows a more negative trend in Arctic sea-ice area than observed over 1979-2014, but there are still 16 models (so 33%), and the multi-model mean (-670,000 km² per decade) has a less negative trend in September sea-ice area than observed (-800,000 km² per decade). If you look at Figure 2 of SIMIP Community (2020, <https://doi.org/10.1029/2019GL086749>), which you mention in your text, you will see that the multi-model mean trend in March Arctic sea-ice area is more negative than observed and the multi-model mean trend in September Arctic sea-ice area is relatively close to observations, except over the very last years (from around 2007, when there is an acceleration of the observed decrease in sea-ice area). As this statement partly motivates your study, I strongly recommend to re-adjust your text and be much more precise than currently by clearly specifying the period, the season, the models considered and the exact region for this specific statement. You also need to make reference to Supplementary Fig. 2c,d, which I suggest to merge with Fig. 1c,d as this is an important

result (you could add CMIP6 results in Fig. 1). For your study, I think it makes more sense to focus only on Barents-Kara Sea and over the winter season for this statement (so no need to mention the total Arctic and summer time).

- We thank the reviewer for your constructive comments on the preciseness of the review of previous studies. In particular, we deeply appreciate the presentation of the reviewer's analysis results. L20-21 refers to the winter sea ice in the Barents-Kara Sea (Fig. 1d; Supplementary Fig. 2), while L36-37 refers to the sea ice in the entire Arctic region. According to the references and the results of the reviewer's analysis, the latter is inaccurate. Therefore, we made two corrections.
- First, we revised the first paragraph to be more accurate, as follows:
 - There is observational and modeling evidence that the retreat of Arctic sea ice has been driven by anthropogenic greenhouse gas emissions¹⁻³, and climate projections using multiple Earth System Models (ESMs) suggest that near ice-free conditions will emerge in the Arctic Ocean in September by the middle of this century⁴⁻⁸. Overall, the reproducibility of the distribution and past variations of Arctic sea ice has much improved in the recent ESM generation^{1-3,8,9}. However, the CMIP6 ESMs still have difficulties in reproducing sea ice in more localized region such as the Barents-Kara Sea (Fig. 1d).
- Second, following the reviewer's comment, we merged Supplementary Figs. 2c and d into Fig. 1c and d.

L58-59: Related to my previous comment, you mention an ‘underestimation of the long-term decrease in sea-ice concentration’. What do you mean exactly? Do you mean that the trend in CMIP6 multi-model mean March Barents-Kara SIC is less negative than observed? If yes, please be more precise and relate it to your Supplementary Fig. 2c,d, which I suggest to merge with Fig. 1c,d as this is an important result. Also, what is SIC here: is it the mean SIC over the entire Barents-Kara region? You define it in the Methods but it would also be good to briefly define it here.

- “Underestimation of the long-term decrease in sea-ice concentration” means that “the less negative trend of DJF Barents-Kara SIC in CMIP6 multi-model mean” (Line 61-62). We have defined the “DJF SIC in the Barents-Kara Sea” in the subsection (“Statistical analysis”) in the Methods. Also, we have mentioned the precise area of “Barents-Kara region” in Line 46-47.

L85-89: I agree that the SIC trend becomes much closer to observations with NAGA

compared to HIST (Fig. 1c,d). However, for SST, as you nudge to COBE-SST2, the NAGA trend logically becomes closer to COBE-SST2 compared to HIST, but it is still $0.05^{\circ}\text{C}/\text{decade}$ off from HadISST2 (Fig. 1a,b). This is due to a relatively large difference in SST trend between the two observational datasets (0.21 vs. $0.24^{\circ}\text{C}/\text{decade}$). Why did you choose COBE-SST2 for the nudging and not HadISST2? You should provide a reasoning for this? Otherwise, one would ask what would happen if you nudged to HadISST2 instead, which has a larger trend in SST. Related to that, you can see that there is a large improvement in SIC trend when using NAGA instead of HIST, relative to observations (which are relatively close to each other). However, the improvement is not that clear for SST trend as the difference between the two observational datasets ($0.03^{\circ}\text{C}/\text{decade}$) is comparable to the difference between NAGA and HIST ($0.02^{\circ}\text{C}/\text{decade}$). I think you should comment on this.

- As the reviewer pointed out, the Gulf Stream SST trend in COBE-SST2 is smaller than in HadISST2. However, due to the computational resource limitation, we had to choose one SST data set. Thus, we selected COBE-SST2 because COBE-SST2 is a “conservative” SST data set compared to HadISST2.
- If we use HadISST2 for SST-nudging, the Gulf Stream SST trend in NAGA will become more than $0.19^{\circ}\text{C}/\text{decade}$. Therefore, we speculate that the ensemble mean of the SIC trend in NAGA may negatively increase up to several per cent larger.
- The difference between HIST and NAGA is based on the ensemble means, i.e., the difference in the externally forced component. On the other hand, the difference between HadISST2 and COBE-SST2 is caused by errors in the data processing. Therefore, it is not appropriate to treat the difference between the SST and SIC trends of the observed data in the same manner as in HIST and NAGA.

6. Data and methodology

L75 and L398-399: Did you start the 10 members of NAGA from 10 different MIROC6 CMIP6 members? If yes, how did you select them? It is not clear from the text.

- We selected different ten member of CMIP6 historical runs by MIROC6 (r1i1p1f1, r2i1p1f1...r10i1p1f1). We have added the following sentence to the Methods section:
 - This experiment includes ten ensemble members corresponding to the initial conditions of each ensemble member in HIST, which are equivalent to CMIP6 historical runs (r1i1p1f1, r2i1p1f1...r10i1p1f1) by MIROC6.

L121: According to Supplementary Fig. 6, this is the horizontal SURFACE heat flux. You should write it. You should also add this precision to the Methods (L440-448).

- We thank the reviewer for pointing out the misleading description. We have mentioned "horizontal subsurface heat flux" in this sentence and Supplementary Figure 6. In addition, the description of Methods has been changed.

L387: A short description on how the 50 MIROC6 members are generated is necessary, as this is the main model used in this study.

- To explain how the 50 MIROC6 members are generated, we have added the following sentences:
 - After 2000-year spin-up, 800-year preindustrial control simulation was conducted⁴⁰. 50 initial conditions were taken from the 800-year simulations with more than 10 years intervals. Using these initial conditions, historical simulations are conducted with the external forcing dataset following the protocol of CMIP6^{ref. 30}.

L388: It should be ‘SSP2-4.5’ instead of ‘SSP245’. Why do you use this specific scenario and not another one? A small justification is needed.

- The reason why we used the SSP2-4.5 scenario is that this scenario is the most moderate. Also, we used the SSP2-4.5 forcing only for the last three years, and thus we believe that the influence of scenario selection is slight. We have added the following sentence:
 - Since the SSP2-4.5 forcing is only used for the last three years, and thus we believe that the impact of the difference between the scenario selection is small in this study.

L390-393: As this is an important method used in your study, you should describe how you nudged SST anomalies to observations. Did you use a correction in the surface heat flux or something else? Do you only nudge at the beginning of the simulation or do you update it through time? Additionally, you need to provide a motivation why you perform this set of experiments with a sentence or two.

- In NAGA, modelled SST anomalies are always restored to the observational SST anomalies with a 3-day relaxation time scale. For SST restoring, the nudging flux is added to the surface heat flux to the ocean. The reason for the applying the pacemaker experiment to the North Atlantic is that we expect that constraining SST anomalies in the Gulf Stream region improve

the response of atmospheric or oceanic heat transport to Gulf Stream SST. To explain the motivation and details of the method of SST restoring, we have added the following sentences to the second paragraph in Methods:

- For SST nudging, the nudging flux is added to the heat flux to the ocean surface:

$$\text{Nudging flux} = \frac{\rho C_p h}{\tau} \times (\text{SST anomaly}_{\text{obs}} - \text{SST anomaly}_{\text{model}}). \quad (1)$$

ρ (=1027 kg m⁻³) is the density of the seawater, C_p (=4187 J kg⁻¹ K⁻¹) is the specific heat of the seawater, τ (= 3 days) is the restoring time scale, and h (=50 m) is assumed to be surface mixed layer affected by SST restoring. Several previous studies adapted this method (so-called pacemaker experiment) to detect the mechanism of climate change (e.g., global warming hiatus⁴²). The reason for the application of pacemaker experiment to the North Atlantic is that we expect that constraining SST in the Gulf Stream region improve the response of atmospheric or oceanic heat transport to the Barents-Kara Sea.

L402-406: What do you exactly mean by ‘constant SST with the magnitude equivalent to a decade of temperature rise from 1970-2017 in COBE-SST2’? What is the purpose of this experiment?

- To explain the setting and purpose of NAGAc more clearly, we have added Figure 5a and explanations to the Methods (Line 510-518). In NAGAc, SST anomalies are always restored to the constant SST anomaly pattern shown in Figure 5a, whose amplitude is equivalent to 10-year SST rise estimated by linear trends of COBE-SST2 over the Gulf Stream region for 1970-2017 (Fig. 5a, Supplementary Fig. 7b).
- As we mentioned in the response to the comment on L138-140 and L402-406, the NAGAc experiment aims to investigate the time evolution of surface temperature and velocity anomalies in the whole North Atlantic against the idealized positive SST anomaly in the Gulf Stream.

L430-439: I understand the logic to randomly select 10 members in HIST to correctly compare to the 10 members of NAGA. But if I understood correctly, the 10 NAGA members were started from 10 different HIST members, is it right? (see my comment L75) If yes, then shouldn’t you simply take the 10 HIST members that correspond to NAGA instead of making a random selection?

- We understand the reviewer’s concern about whether we should simply use 50 HIST members or not. The reason for using the 50 HIST members is to highlight that the

underestimation of the ensemble-mean SST and SIC trends in HIST is not due to the ensemble size. We used the 50 HIST members directly when we plot time-series scatters of Gulf Stream SST and Barents-Kara SIC trends (scatters in Fig. 1b,d, Fig. 6b). For example, Figure 1d shows that the observed SIC trend is not captured even by HIST 50 members. Thus, we need to use 50 HIST members at least in this analysis.

- For the other analyses, we used only the 10 HIST members that correspond to NAGA. The first manuscript was not clear about the number of members, and thus in revised manuscript we have added the following sentences to the Methods:
 - Note that this Monte Carlo method is used to calculate the time series and the box-plot in Fig. 1 and scatter plots in Fig. 6. For the other analyses, only 10 HIST members that correspond to NAGA are used.

L461: A description of observational datasets (COBE-SST2 and HadISST2) is missing in the Methods. Also I didn't find a definition of the acronym COBE-SST2 in your text.

- We have added the description of COBE-SST2 and HadISST2 to the beginning of the Methods. A definition of the acronym COBE-SST2 is also shown as follows:
 - For monthly SST and SIC, Centennial In Situ Observation-Based Estimates of the Variability of SST and Marine Meteorological Variables version2 (COBE-SST2)³⁸ and Hadley Centre Sea Ice and Sea Surface Temperature version 2 (HadISST2)³⁹ are used for 1970-2017.

L615-618: EC-Earth3 also has more than 10 members for the historical CMIP6 simulations. Why didn't you use this model?

- When we downloaded the CMIP6 historical dataset, we could not get the EC-Earth3 due to the server error. We have added 25 members of EC-Earth3 to the analysis which could be downloaded when we tried again.

7. Clarity and context

L33-36: This sentence is confusing as it mixes both future model projections of Arctic sea ice (in the first part of the sentence) and recent observations (in the second part of the sentence). I think it would be better to state that 'there is observational and modeling evidence that the retreat of Arctic sea ice has been driven by anthropogenic greenhouse gas emissions'. Also, the references you provide are a bit outdated and there have been more recent studies related to this, e.g. Notz & Stroeve (2016,

<https://doi.org/10.1126/science.aag2345>) and Stroeve & Notz (2018, <https://doi.org/10.1088/1748-9326/aade56>).

- We thank the reviewer's constructive suggestion. We rephrased this sentence to "There is observational and modeling evidence that the retreat of Arctic sea ice has been driven by anthropogenic greenhouse gas emissions". Also, we refer Notz and Stroeve (2016) and Stroeve and Notz (2018) instead of Stroeve et al. (2007) and Comiso et al (2008).

L46 and other instances in the text: You always write 'Barents-Kara sea-ice variability', but this is not precise. You should rather say 'Barents-Kara sea-ice area (or extent or concentration) variability'.

- We thank the reviewer's helpful comment on the preciseness. In most cases, 'Barents-Kara sea-ice' means 'Barents-Kara sea-ice concentration' although it depends on the context. We corrected this line and others in the manuscript to be more precise. (Line 46, and others)

L48: 'the winter sea-ice tends to be less...': less what? Don't you mean that the winter sea-ice concentration or area tends to be lower?

- We mean that "the winter sea-ice area tends to be lower than usual". We have rephrased this sentence.

L62-63: As it is, the reader can have the feeling that it is the study of Kosaka & Xie (2013) that demonstrated that the pacemaker experiment provides better results than CMIP6 historical runs. I suggest you remove the reference '30' to Kosaka & Xie (2013), which is a very different study, and re-write: 'Our pacemaker experiment demonstrates that the Barents-Kara...' Also, it is not clear what you mean by SIC: is it the mean SIC over the whole Barents-Kara Sea? Wouldn't it be better to use sea-ice area or extent instead?

- We thank the reviewer's helpful comment. We agree with the reviewer that this sentence is misleading and decide to move the reference from this sentence to the Methods section to explain the history of pacemaker experiments (second paragraph of "Historical (HIST) and North Atlantic-Global Atmosphere (NAGA) experiments") (Line 503-505).
- Also, "Barents-Kara SIC" means that "SIC area-averaged over the whole Barents-Kara Sea". We corrected this sentence to be more precise. The reason why we use SIC is that SIC is a good indicator of sea-ice variability. Figure R1 shows the time-series of SIC, sea-ice area,

and sea-ice volume on the entire Barents-Kara region for HIST and NAGA. All time-series in NAGA decrease faster than HIST and show the qualitatively same result. Since it is the easiest to calculate the SIC time-series, we selected SIC for analysis in this manuscript.

Figure. R1 | Simulated time series and linear trends of DJF sea-ice in the Barents-Kara Sea from 1970–2017. a. DJF mean SIC (solid line) and linear trends (dashed line) averaged over the Barents-Kara Sea for the ensemble means of 10 members for HIST (blue) and NAGA (red). Shading indicates one standard deviation of ensemble members in both experiments. The linear trend per decade for each data set is shown in the legend. **b,c.** As in **a**, but for **(b)** sea-ice area and **(c)** sea-ice volume.

L63: You miss something after ‘can be better reproduced’. I guess you mean that the SIC trend can be better reproduced if SST anomalies in the Gulf Stream region are constrained by observations.

- We mean “...can be better reproduced if SST anomalies in the Gulf Stream region are constrained by observations”. However, since this is a prolix sentence, we have changed this to the following sentence:
 - This pacemaker experiment can reproduce the SIC trend averaged over the entire Barents-Kara Sea from 1970–2017 better than the CMIP6 historical simulation products.

L69: I wouldn’t use an acronym (NAGA) in the section title, but rather the full name for clarity.

- We have used “North Atlantic–Global Atmosphere” in this section title.

L71: You need to add that MIROC6 is a global climate model, especially for the audience of Nature Communications.

- We added a sentence to explain that MIROC6 is one of global climate models participating CMIP6.

L73: ‘The other set of experiments is...’

- We have corrected this sentence.

L74: What do you mean by ‘strongly restored’? Is ‘strongly’ necessary?

- “Strongly restored” means nudging time scale is shorter than the other pacemaker experiments. However, we agree that “strongly” does not need here, and we have removed.

L77: Please write the months taken into account for the winter average: is it DJF?

- We have added “December-January-February; DJF” to this sentence.

L87: In fact, the NAGA trend is lower than HIST (as there is a minus sign)... You should instead write ‘more negative’ or ‘larger in absolute value’.

- We have changed “larger” to “more negative”.

L92: As this metric is important, you need to better describe what is the ‘ensemble mean variance’ and what is the ‘total variance in the ensemble’.

- To explain this metric, we have added the following paragraphs as a subsection (“Ratio of the ensemble mean variance to all variances”) to the Methods as follows:

- **Ratio of the ensemble mean variance to all variances**

The ratio of ensemble-mean variance to the total variance in the ensemble for X is calculated as below:

$$\text{variance ratio} = \frac{1}{n} \sum_i^n (X_{\text{enmean},i} - \overline{X_{\text{enmean}}})^2 \bigg/ \frac{1}{N} \sum_j^{10} \sum_i^n (X_{j,i} - \overline{X}_j)^2 \quad (2)$$

Here, n is the length of the time, X_i is X for t=i, X_{enmean} is the ensemble mean, \overline{X} is the time mean of X, $X_{j,i}$ is X for ensemble member j and t=i, and N (=n × 10) is the total time length for 10 members. We refer to the numerator (denominator) of Eq. (2) as ensemble-mean variance (total variance in all ensembles). This is analogous to the signal-to-noise ratio. When

the variance ratio is 1, variations for all members are fully constrained to the ensemble mean.

L105-106: What are ‘linear trends of oceanic states’?

- We corrected this sentence to “linear trends of surface ocean and lower atmosphere”.

L130-132: Please be more precise: ‘In NAGA, both the velocity and temperature trends (vertically-averaged over the first 200 m) in the Gulf Stream region and Norwegian Sea are larger than those in HIST. Also, the warming in the Gulf Stream...’

- We have corrected this sentence to be more precise.

L133-134: The Gulf Stream slowdown you mention is confusing as you previously talk about larger temperature and velocity in the Gulf Stream region in NAGA (L130-131). I think you want to say that there is a small spot of cooling at about 40°W / 45°N, and also the width of the Gulf Stream in the eastern branch seems to weaken in NAGA compared to HIST. Is that correct? If yes, please be more precise.

- We mean “amplitudes of the velocity and temperature trends in NAGA are larger than those in HIST”. We have also added the velocity trends in the revised manuscript (Fig. 4a,b). In these figures, it is found that the Gulf Stream is clearly decelerating (and accelerating in a small region off the eastern coast of the U.S.) in NAGA. Although a similar deceleration pattern can be seen in HIST, the amplitude is much smaller than in NAGA.

L151: I wouldn’t use the SIC acronym (but rather the full name instead) in the section title for the clarity of the paper.

- We have changed SIC to sea-ice concentration.

L161: You need to make reference to Supplementary Table 2.

- We have added Supplementary Table 2 for the reference.

L173: What do you mean by ‘the variance of the SIC trend’?

- This means the variation of the SIC trends between ensemble members. Thus, we have added

“between ensemble members”.

L177: I don't clearly understand what you did when you say that you removed the multi-model mean trend.

- To capture the dominant mode of the inter-model variations, multi-model mean has been removed before the EOF analysis. To make it more clearly, we have changed this sentence to “after the multimodel mean are removed to capture the inter-model anomalies”.

L217-218: For Fig. 1a,c, I suggest to write SST and SIC ‘anomalies’, respectively, in the Y axis label.

- We have added ‘anomalies’ to Y axis labels.

L221: There are several lines in Fig. 1a,c, so please use the plural. Also, linear trends in Fig. 1a,c are not shown by dashed lines but rather by thick solid lines. But for the clarity of your figure, I would recommend using dashed lines, as you state in the caption.

- We agree the reviewer and have used linear trends with dashed lines.

L225: This is not the ‘number of linear trends’, this is rather the trend in SST / SIC for each dataset.

- We have corrected “number of linear trends per decade” for “Linear trend per decade for each data set”.

L244: This is not the surface temperature and velocity but rather vertically-averaged temperature and velocity in the first 198 m.

- We have decided to show the potential temperature and horizontal velocity vertically-averaged within the surface mixed layer in Fig. 4. Thus, we have changed this sentence to “Linear trends of the potential temperature and horizontal velocity vertically-averaged within the surface mixed layer in the North Atlantic for NAGA and HIST”.

L393-394: You should be more explicit and make reference to Supplementary Fig. 3b as this

is where the solid box is.

- We added the precise definition of the solid and dashed box and made a reference to Supplementary Fig. 3b here.

L408: Shouldn't it be 31 December 2017 instead of 1979?

- NAGAc experiment started from 1970 and integrated for 10-years. Thus, the end of the integration is 31 December 1979.

L521-524: The two first sentences of the caption of Supplementary Fig. 3 are redundant. Please consider merging them together. Also, it is not clear what you mean by 'ensemble mean variance' and 'total ensemble member variance' (also present at L92). Which period is considered here?

- We merged these sentences. Also, as we mentioned in the reply to the comment on L92, we have added the explanation to the Methods to explain the meaning of the metric. To make this figure easy to understand visually, we have also changed colors.

L536: The unit '°C' is missing for 'potential temperature [°C decade-1]'.

- We have corrected the unit. This figure has been modified and moved to Fig. 3 in revised version.

L553: In order to make this figure visually more attractive, I would color the map with the magnitude of surface heat flux. You can let the vectors showing the direction of surface heat fluxes.

- Following the reviewer's comment, we show the magnitude of the surface horizontal heat flux trend in color (Supplementary fig. 6). We thank the reviewer's constructive suggestion, which makes this figure more attractive and informative.

L567: This is a nice figure but the legend text could be enlarged for clarity. Maybe you could include this figure as a panel of Fig. 4 to better support your conclusions.

- Since the legend text was too small, we enlarged the text. Also, following the reviewer's

suggestion, this figure has been merged to Figure 6.

8. References

L46-56: You mention two main drivers of Barents-Kara SIC at interannual time scales, namely ocean heat transport and Gulf Stream SST meridional shift. In your study, you make the hypothesis that it is in fact a combination of the two that is responsible for the sea-ice loss. However, in this part, you only say that the Gulf Stream meridional shift could enhance the atmospheric heat transport and you provide references for this. What about references that make a connection between Gulf Stream SST and ocean heat transport? If they exist, they should be cited here. If not, I think it should also be said.

- For example, Nakanowatari et al. (2014) (ref 26) explicitly mentioned that water temperature anomalies from the North Atlantic subpolar gyre are related to interannual variability of Barents-Kara sea-ice. This has been mentioned in this paragraph. However, in our current knowledge, the direct link between the Gulf stream SST itself and Barents-Kara SIC variability through ocean heat transport has not been discussed. To note this point, we have added the following sentence:
 - Although the direct impact of heat transport from the Gulf Stream on Barents-Kara Sea is still under debate, ...

L71: You need to add a reference for MIROC6, even if done in the Methods.

- We referred Tatebe et al. (2018) here.

L73: You need to add a reference to the CMIP6 description paper (Eyring et al., 2016, <https://doi.org/10.5194/gmd-9-1937-2016>).

- We have referred Eyring et al. (2016) here for a reference to CMIP6.

L109: As this is your result, I wouldn't provide a reference to Arthun et al. (2012) here.

- We have deleted this reference.

L283: You should revise the reference to SIMIP Community: it should be 'SIMIP Community (2020)...' and not 'Community, S. I. M. I. P. (2020)...'

- We have corrected the reference.

Reviewer #2

This study presented the impact of Gulf Stream warming on sea ice decline over the Barents-Kara Seas in winter using ensemble climate models. Authors found that there is a negative relationship between linear trends of upper layer ocean temperature over the Gulf Stream and sea ice concentration over the Barents-Kara Seas from 1970 to 2017. The Gulf Stream warming slowly propagates northeastward along the European region and reaches the Barents-Kara sector, inducing sea ice decline over the Barents-Kara Seas. To my knowledge, this is a new study to show link between Atlantic ocean variability and Arctic ocean climate. It will be a good piece of work for better understanding Northern Hemisphere climate variability. However, there are a few things that could be done to strengthen this paper.

- We would like to thank the reviewer's time and effort in reviewing our manuscript. The reviewer's constructive comments encouraged us to conduct additional analysis to highlight the role of the atmosphere and ocean dynamics on the Barents-Kara sea-ice. We hope our revised manuscript satisfies all reviewer's concerns. The reply for each comment is shown below.

It is not enough for discussing the impact of atmospheric circulation variability over the Northern Hemisphere on sea ice decrease over the Barents-Kara Seas. Previous studies reported that the atmospheric response to warming over the Gulf Stream has direct and/or indirect impact on Arctic sea ice decline. In addition, over the Barents-Kara sector, northward movement of sea ice is associated with anomalous southerly winds reported by previous studies. I think authors should discuss results of linear trends of atmospheric circulations.

- We thank the reviewer for your constructive comment and suggestion. Following the suggestion, we have conducted additional analyses to examine the linear trends of the lower atmospheric circulation over the Barents-Kara Sea.
- To clarify the importance of the atmospheric circulation variability on the Barents-Kara SIC reduction, we have investigated linear trends of lower atmospheric circulation and temperature (Fig. 2c, d). This analysis has revealed the minor role of the atmospheric pathway from the Gulf Stream to the Barents-Kara Sea compared to the oceanic pathway. To explain this result, we have added the following paragraph in the revised manuscript (Line 114-129):

- Differences in surface wind trends appear to cause sea ice to retreat more poleward in NAGA, but the trends are not statistically significant (Fig. 2c,d). Moreover, sea-ice drift trends in NAGA and HIST respond to surface ocean circulation changes rather than wind changes (Fig. 2a-f), suggesting the differences in the dynamical contribution of the atmosphere to sea-ice loss are not the cause of the sea-ice loss difference between NAGA and HIST. While, the spatial agreement of negative trends in NAGA between the SIC, surface salinity, and surface heat fluxes (Supplementary Fig. 4a,c) means that the atmospheric surface warming trend over that region (Fig. 2c) is the result of sea-ice loss rather than the cause. The same is true in the difference between NAGA and HIST. The more considerable SIC decrease in NAGA is not due to the difference in atmospheric heat advection, which is different in the case of interannual variability²⁶. The heat release difference is due to the less sea-ice formation (i.e., the less SIC), which is implied by the more negative trends of surface salinity flux in NAGA (Supplementary Fig. 4c,d). Therefore, the improved SIC trend in the Barents-Kara Sea results from the SST warming (Fig. 2e, f), possibly driven by oceanic heat transport from the North Atlantic domain.

From difference in upper-level ocean temperature between NAGAc and HIST, the Barents-Kara Seas warming is clear after 7 years, indicating warm ocean temperature anomaly reaches Barents-Kara Seas sectors. However, from the annual mean horizontal velocity in figure 3, it is unclear to me that warming temperature anomaly propagates northeastward along the European coast after 3 years later, in particular in 4 and 5 years. I recommend authors show northward advection of warm temperature anomaly from subtropical gyre region to the European coast region clearly.

- We appreciate the reviewer's constructive comment, which helped us thoroughly revise the analysis on the ocean heat transport in the NAGA and NAGAc. To show the result in NAGAc more clearly, we have added the new figure (Fig. 5). This new figure demonstrates that surface temperature warming in the Gulf Stream region gradually extends northeastward, and it makes a significant difference from HIST in the Barents-Kara Sea after about seven years.
- First, to show the time evolution of differences in the ocean surface temperature and velocity in NAGAc and HIST, we have calculated the time series of water temperature in four regions (Fig. 5b,c). In NAGAc, the temperature increases from the Gulf Stream region to the northeast, suggesting more heat is transported from the Gulf Stream region to the pole.
- Second, we have also shown snapshots of differences between NAGAc and HIST in water

temperature and velocity from year 1 to 9 (Fig. 5d-l). From year-1 to 3, the temperature differences gradually develop around the Gulf Stream region (Fig. 5d-f). In the year-4 to 5, positive water temperature differences also develop off the coast of Portugal and the UK (Fig. 5g,h). This temperature difference seems to extend northward along the Norwegian coast and toward the Barents Sea (Fig. 5i-l).

Specific comments:

1) To discuss the impact of surface atmospheric parameters on sea ice concentrations, authors should show linear trends of atmospheric parameters (e.g. sea level pressure, temperature, and wind speed in the lower troposphere). The geopotential height at upper troposphere (e.g. 300hPa) would be good for discussing the teleconnection between mid- to high-latitudes.

- We thank the reviewer's constructive comment on the atmospheric impact on the Barents-Kara sea-ice reduction. As we mentioned in reply to the first major comment, we have investigated linear trends of lower atmospheric circulation over the Barents-Kara Sea (Fig. 2c,d). We conclude that the improved SIC trend in the Barents-Kara Sea results from the SST warming, possibly driven by oceanic heat transport from the North Atlantic domain.
- We have also checked the linear trends of geopotential height at 300 hPa and temperature at 950 hPa in the northern hemisphere, as in the previous study (Sato et al. 2014; ref. 26) (Fig. R2). The results show that there is no wave-like atmospheric circulation trend similar to the teleconnection from the Gulf Stream.

Figure R2. | Linear trends of temperature and geopotential height in the northern hemisphere. Linear trends of DJF mean temperature at 950hPa [$^{\circ}\text{C decade}^{-1}$] for (a) NAGA and (b) the difference (NAGA minus HIST). c and d As in a and b, but for geopotential height at 300hPa [m decade^{-1}].

2) If we focus on warming ocean temperature anomaly during 7 years in NAGAc, this warming anomaly seems to move northeastward from the subtropical gyre region to the Norwegian Sea via the European coast. However, to my knowledge, the northward movement of ocean current is weak over the subtropical gyre, suggesting that ocean current would have a small impact on the propagation of warm temperature anomaly. The northward movement of ocean current from the subtropical gyre to the European coast region is clearly seen in the climate model? or there is another mechanism for this propagation without ocean current?

- To answer the reviewer's question, we have analyzed the circulation changes in the North Atlantic in NAGA and HIST. It is found that most surface ocean current branches weaken especially in NAGA, but only northward current from the eastern edge of the nudging region to the Norwegian Sea strengthened (Fig. 4a,b).

- Furthermore, we have conducted the mixed layer heat budget analysis to investigate the relative role of the ocean and atmosphere on surface ocean warming. This new analysis revealed that the oceanic contribution in NAGA is larger than HIST and extends to the downstream of the Gulf Stream, which corresponds to the larger temperature warming in the Norwegian Sea (Fig. 4e,f). To show this result, we have added the following paragraph (Line 164-169):

- In both HIST and NAGA, ocean dynamics contribute to surface temperature warming in the Norwegian Sea (Fig. 4e,f), but its contribution is larger in NAGA and extends to the downstream of the Gulf Stream (around 30°W and 50°N), which is consistent to the larger temperature warming in the Norwegian Sea (Fig. 4c,d). The positive ocean contribution trend is found along the acceleration trends of surface velocity, suggesting that poleward ocean heat transport strengthens.

- As we mentioned in reply to the second major comment, the time evolution of differences between NAGAc and HIST also suggest positive surface temperature anomalies gradually extend poleward from the downstream region of the Gulf Stream (Fig. 5). Therefore, northward movement by ocean current is found at least from the downstream of the Gulf Stream.

3) I am not sure the Gulf Stream warming anomaly reaches the Barents-Kara sea ice decreasing trend area (eastward of 50E). To show advection of warming anomaly to sea ice decline area clearly, I would like to see the extended areas over the Barents-Kara sea (e.g extended to 60E).

- Significant linear trends of ocean temperature anomalies and current velocities are also found around 60°E (Fig. 2e,f). The atmosphere does not lead to surface water warming in the eastern Barents Sea because linear trends of the surface heat flux indicate increased heat release from the ocean (Supplementary Fig. 4a,b). Therefore, we cannot explain the surface ocean warming in the eastern Barents Sea without ocean dynamics. Since the surface ocean warming in the Barents-Kara Sea is most likely due to the heat transport from the North Atlantic domain, these results imply that the Gulf Stream warming anomaly should reach the Barents-Kara sea-ice decreasing trend area.

4) The horizontal heat flux and ocean temperature at a depth of 54m were shown in Figure 2. However, the sea ice is influence by ocean temperature and current near-surface. Therefore, authors should show averaged horizontal heat flux and ocean temperature from the surface to a depth of several tens or several hundred such as figure 3, or show vertical

distributions of potential temperature with horizontal heat flux over sea ice decreasing trend area such as Supplementary Figure 4a-c (e.g. 50E, 70N to 85N).

- The surface ocean velocity and temperature trends have been added to the figure (Fig. 2e,f) because sea-ice in the model is affected by surface water temperature and velocity in the first layer of the ocean model. Also, since water temperature and horizontal heat transport around 80°N are uniform in depth (Fig. 3a,b), the horizontal heat flux at 54m depth is representative of the heat transport in all depths. Thus, we have decided to show the horizontal heat flux trend at 54m to examine the horizontal heat flux in the Barents-Kara Sea and the Norwegian Sea (Supplementary Fig. 6).

5) Author conducted statistical analysis for figures 2 and 3? I think it is important that upper-level ocean temperature in NAGAc has statistically significant warming. Please show areas with statistically significant trends.

- We appreciate the reviewer's constructive suggestion. We performed the modified Mann-Kendall test for the statistical test of linear trends (Methods). This statistical test highlights the role of the ocean contribution on the surface ocean warming in the Barents-Kara Sea (Fig. 2) and North Atlantic (Fig. 4).

Reviewers' Comments:

Reviewer #1:

Remarks to the Author:

Please see attached file.

Second Review of

'Barents-Kara sea-ice decline attributed to surface warming in the Gulf Stream'

By Y. Yamagami *et al.*

Submitted to *Nature Communications*

1. General comment

I congratulate the authors for having taken into account all my comments with great care. I think the paper is now much more powerful and the key results are better backed up by evidence. In particular, I am now much more convinced by the ocean pathway linking the Gulf Stream SST to the Barents-Kara SIC, clearly evidenced by Figs. 2 to 5 and including an additional mixed layer heat budget analysis. I also congratulate the authors for having clarified many methodological points. The paper is now much clearer and will greatly benefit the scientific community.

I have a couple of minor comments / corrections below, which I think would improve the paper even more. I recommend the paper for publication and I do not need to see a revised version of it.

2. Minor comments

L19: Remove 's' in 'concentrations' and 'SICs'; it should be 'sea-ice concentration (SIC)'.

L32: I would say 'the recent retreat of Arctic sea ice'.

L34 and elsewhere in the paper: I recommend to use the term 'global climate models (GCMs)' instead of 'Earth system models (ESMs)', as many of the CMIP6 models do not simulate chemical and biological processes, which are simulated by ESMs. So an ESM is a GCM, but a GCM is not an ESM.

L40-41: Replace 'when sea ice is most productive' by 'when sea-ice formation is largest'.

L46: Replace 'decrease trend' by 'negative trend'.

L46: Remove coma after 'and'.

L66: 'can reproduce' (without 's').

L74-75: Remove 'ref.' before '29' and before '30'.

L111: Is 'First' really needed? There is no 'second' and I don't really see the usefulness of this word here.

L111-112: 'linear trends of surface ocean and lower atmosphere' does not mean anything. Didn't you miss the word 'variables'? ('linear trends of surface ocean and lower atmosphere variables')

L114: Do you mean the differences between NAGA and HIST? Please be more precise.

L115: How can you see in Fig. 2c,d that the trends in surface wind are not significant since they are represented as vectors?

L117-119: I suggest to rephrase more positively into: 'suggesting the difference in sea-ice loss between NAGA and HIST has an oceanic origin rather than atmospheric'.

L119-122: How does the spatial agreement between SIC, surface salinity and surface heat flux show that the atmospheric surface warming is a result of sea-ice loss? You need to better explain.

L123: 'The larger SIC decrease' instead of 'The more considerable SIC decrease'.

L123-125: Is this sentence really necessary? If yes, you need to rephrase it as it is barely understandable.

L125-126: Replace the two 'less' by 'lower', which is more appropriate.

L127-129: I guess you mean the improved SIC trend in NAGA compared to HIST? Also, I guess you mean 'SST warming in NAGA compared to HIST'? Please be more precise here.

L132-138: I think your explanation of Fig. 3 is a bit confusing here as you insist a lot on the large difference in the northern part of the BSO section, while differences between NAGA and HIST are larger at 65°N and 70°N. Also, as explained in my comment for Fig. 3 below, I don't really see the point to show both transects at 65°N and 70°N as they show a relatively similar behavior. I would remove the transect at 65°N from the paper. I would rephrase: 'We find that the poleward heat transport from the North Atlantic (measured at two different transects) increases more in NAGA than HIST, which leads to the faster reduction of Barents-Kara SIC in NAGA. This increase is larger at 70°N compared to the Barents Sea Opening section, where the increase is localized to the northern part of the section (Fig. 3).'

L142-143: are investigated' (plural).

L144-145: Replace 'The large difference' by 'A large difference'. Also, I think you mean '65°N' instead of '75°N'.

L149-152: Are you sure you are not mixing temperature-induced and velocity-induced trends here? According to Supplementary Fig. 6e-f, it is the contribution of the temperature-induced trend that is more important in the center of the Barents Sea. West of the Barents Sea Opening Section, the contribution of the velocity-induced trend is larger (Supplementary Fig. 6c-d). Please make sure your statement is correct here.

L151: 'temperature-induced trend' (remove 's').

L157-160: I would place reference to Fig. 4 at the end of this sentence.

L159: Replace 'in the upstream from the Norwegian to the Gulf Stream' by 'in the region between the Gulf Stream and the Norwegian Sea'.

L167: 'which is consistent with the larger...'

L174: 'induce' instead of 'induces'.

L183: 'the rest accumulates'.

L217-218: 'the larger the signal in a model, the larger...' (remove 'is').

L219: 'When a similar analysis...'

L224: 'a similar analysis'.

L239: 'with those studies' (remove 'of').

L244-245: I would replace by 'This relationship is found both in a multi-member analysis performed with MIROC6 and in the CMIP6 multi-model ensemble'.

L245-247: 'Although a connection..., our results go beyond this simple relationship and show that...'

L250: 'The model underestimation...'

L288: Replace 'transections' by 'transects'.

L480-486: You mix present and past tenses. Please be coherent and choose one or the other.

L480: 'After a 2000-year spin-up, a 800-year preindustrial...'

L482: 'simulation' (without s).

L484: Remove 'ref.' before '30'.

L487: 'the most moderate scenario' is not really a good terminology. What does 'moderate' mean? I recommend to say 'a middle-of-the-road greenhouse gas emission scenario'.

L488: Remove 'Since' in order to have a correct sentence from a grammatical point of view.

L492: Related to one of my previous comments (Validity, L85-89) and your reply, I think you should add in the Methods why you chose COBE-SST2 for the nudging and not HadISST2. Your reply 'we selected COBE-SST2 because COBE-SST2 is a "conservative" SST data set compared to HadISST2' is not convincing to me. What do you mean by "conservative"?

L497-498: In order to define all terms in equation (1), I would say: ‘...in the historical simulations (**SST anomaly**_{model}) and COBE-SST2 (**SST anomaly**_{obs}), respectively’.

L612-614: In the world in which we live, it is always better to make the codes available in a public repository, so I would encourage the authors to do so. This will make your study even more visible.

Fig. 1c,d: I think it was clearer in the first version of the paper to have empty dots (blue, red and green dots) instead of filled dots, as it is now. This way, you could better distinguish the different dots. Also, I would make sure to clearly see the labels ‘HadISST2’ and ‘COBE-SST2’ in these two panels, as they are not well placed.

Fig. 2b,d,f: For the maps of NAGA-HIST, do you show the difference between NAGA and HIST for the vectors or is it the NAGA value?

Fig. 3: While I understand the logic to have one transect in the BSO section and one at 65°N or 70°N, I don’t understand why you need to show both 65°N and 70°N, which are relatively similar. Personally, I would remove the transect at 65°N and only show 6 panels (instead of 9) in Fig. 3.

Fig. 3: I think you’re missing labels for contour lines of potential temperature trends.

Fig. 4: The black contours for velocity trends are a bit useless if we can’t identify the values of those on the maps. I would remove them, except if you can add the values on the maps.

Fig. 6a: In the title of this panel, remove ‘s’ as it should be ‘one member’.

Fig. 6a: I think there is a problem with the box-and-whisker plot of SIC trend as the black line for ensemble mean is not aligned with the green line in the main plot (corresponding to the green star).

Supplementary Fig. 5: Add ‘ocean’ before ‘heat content’: ‘ocean heat content’.

Reviewer #2:

Remarks to the Author:

Review of NCOMMS-21-24661A

Barents-Kara sea-ice decline attributed to surface warming in the Gulf Stream
by Yamagami, Y., Watanabe, M., Mori, M. & Ono, J.

In revised version, I think that additional analyses solved my questions. The minor role of atmospheric circulation trends on Barents-Kara SIC decrease were shown in Fig. 2. I think that the poleward transport of warming ocean from the North Atlantic sector to Barents-Kara Seas sector is clear in Fig. 5. Therefore, I have some minor comments in revised version.

Minor comments:

- Lines 118-119: "are not the cause" may be too strong message. Because wind speed trend would contribute to poleward shift of sea ice edge over the west part of Barents Sea even without statistically significant. I think that "are not the major cause" is better.

- Line 142: Not 'is', maybe 'are'.

- All figures : For reader, please add the units of color bar like as Figure 4.

- Figure 3: Please show the statistically significant area for horizontal heat transports over each section using symbol (e.g. hatching).

- Caption of Fig. 3: Please add the more information for gray contours (i.e. contour interval, dashed lines mean negative value?).

- Caption of Fig. 4: Please add the sentence for hatching and vector colors like as Fig.2.